# ADAPTIVE KNOWLEDGE TRANSFER FOR GENERALIZED CATEGORY DISCOVERY

## ABSTRACT

We tackle the general category discovery problem, which aims to discover novel classes in unlabeled datasets by leveraging the information of known classes. Most previous works transfer knowledge implicitly from known classes to novel ones through shared representation spaces. However, the implicit nature of knowledge transfer in these methods poses difficulties in controlling the flow of information between known and novel classes. Furthermore, it is susceptible to the label uncertainty of unlabeled data learning. To overcome these limitations, our work introduces an explicit and adaptive knowledge transfer framework that can facilitate novel class discovery. This framework can be dissected into three primary steps. The initial step entails obtaining representations of known class knowledge. This is achieved through a pre-trained known-class model. The subsequent step is to transform the knowledge representation to enable more targeted knowledge transfer, realized through an adapter layer and a channel selection matrix. The final step is knowledge distillation, where we maximize the mutual information between two representation spaces. Furthermore, we introduce a challenge benchmark iNat21 which is comprised of three distinct difficulty levels. We conduct extensive experiments on various benchmark datasets and the results demonstrate the superiority of our approach over the previous state-of-the-art methods.

## 1 INTRODUCTION

Despite the notable achievements of various deep learning models (He et al., 2016; Dosovitskiy et al., 2020), they still face challenges in open-world scenarios when encountering novel concepts. In contrast, humans are able to leverage their existing knowledge for the discovery of new concepts. For instance, when presented with data belonging to previously unseen classes, humans can effectively cluster them based on their prior knowledge. Taking inspiration from this capability, Han et al. (2019; 2021) introduce the concept of the novel class discovery (NCD) problem, while Vaze et al. (2022a) propose a more practical setting named generalized category discovery (GCD), which is our focus in this work. The key challenge of GCD is to effectively utilize the rich semantic information inherent in known classes to facilitate the discovery of novel classes.

In order to discover novel classes, most GCD works (Vaze et al., 2022a; Wen et al., 2022; Zhang et al., 2022; Sun & Li, 2022) follow a single-stage training paradigm in which all data, whether labeled or unlabeled, is amalgamated into a unified learning process with a shared encoder. This entails an implicit manner of knowledge transfer in these approaches. However, such an implicit strategy makes it challenging to regulate the flow of information between labeled and unlabeled data, and can be susceptible to the label uncertainty of unlabeled data in the learning process.

Recent work has attempted to explore explicit strategies for knowledge transfer in NCD, outperforming the traditional implicit strategies (Gu et al., 2023). Nonetheless, they typically focus on distilling class relations in the model output space and hence are limited in the scope of transferred knowledge. On the other hand, we observe that the representation of a trained model for known classes can also provide highly informative clustering cues for the novel classes, resulting in competitive class discovery performance (as shown in Tab.1). This naturally raises a question on the form of knowledge transfer: What is the most effective explicit transfer strategy for the GCD problem?

Building upon this motivation, we design an explicit knowledge transfer paradigm as shown in Fig.1, comprising three core steps: 1) **Knowledge generation**. The initial step learns a representation of

| Method | BL | Clu. |
|--------|------|------|
| ImgNet100 | 78.7 | 71.7 |
| CUB | 58.5 | 62.7 |
| Scars | 42.0 | 41.5 |
| Aircraft | 46.2 | 42.5 |

Table 1: "BL" denotes the typical GCD method. "Clu." denotes clustering novel classes on the pre-trained known-class model without joint training. Details in Sec.4.3.

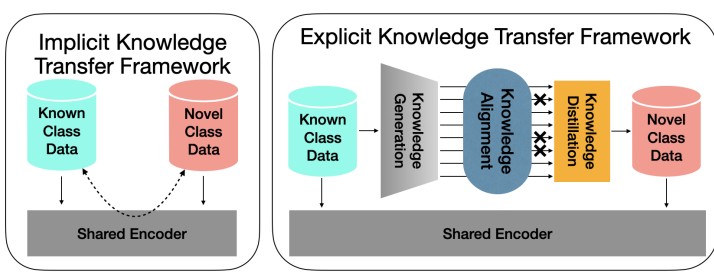

Figure 1: The comparison of our explicit knowledge transfer framework and implicit knowledge transfer framework.

the known-class knowledge from the labeled data. 2) **Knowledge alignment**. After obtaining the known-class representation, we introduce an alignment step that adaptively transforms the initial representation to facilitate the novel class discovery. In particular, we incorporate a nonlinear feature transform with a data-specific channel-wise selection mechanism, which allows us to keep relevant information in an efficient manner for the classification tasks of known and novel classes. 3) **Knowledge distillation**. Armed with the aligned known-class knowledge representation, the third step is to distill this knowledge to facilitate the learning process of known and novel classes using the unlabeled data.

To achieve this, we develop a novel GCD framework that instantiates the above three essential steps. Specifically, we first train a deep network model on the labeled known-class data and use it as our knowledge representation. After that, we employ an *Adapter Layer* to transform the pretrained feature representation so that it is better aligned with the joint representation of all classes. Subsequently, our method introduces a *Channel Selection Matrix* to choose relevant feature channels for the novel class learning. Finally, we propose a new contrastive loss for learning the knowledge transfer from the known to novel classes, which maximizes the mutual information between the transformed known-class representation and the joint feature representation used for classifying both known and novel classes. To cope with the mixed nature of the unlabeled data, we also adopt a mixed-up negative sample generation scheme in the contrastive learning.

We conduct extensive experiments on six widely-used benchmarks and our method achieves significant improvement over the previous state of the art, which showcases the efficacy of our method. The experimental analysis further demonstrates the effectiveness of each module in our design. Our contributions can be summarized as follows:

- We propose a novel explicit knowledge transfer framework for generalized category discovery, which can transfer known-class knowledge more effectively to novel class learning.

- We develop an adapter layer and a channel selection matrix for better knowledge alignment, and a new contrastive loss to encourage the knowledge transfer in model learning.

- We conduct extensive experiments on several benchmarks to validate the effectiveness of our method, which outperforms the SOTA by a significant margin. Particularly, we introduce iNat21, a new benchmark with three difficulty levels, to assess our framework's performance.

## 2 RELATED WORK

**Novel Class Discovery.** The problem of NCD is formalized in (Han et al., 2019), aiming to cluster novel classes by transferring knowledge from labeled known classes. Specifically, KCL (Hsu et al., 2018a) and MCL (Hsu et al., 2018b) use the labeled data to learn a network that can predict the pairwise similarity between two samples and use the network to cluster the unlabeled data. Instead of using pairwise similarity to cluster, DTC (Han et al., 2019) utilizes the deep embedding clustering method (Xie et al., 2016) to cluster the novel class data. Later works mostly focus on improving the pairwise similarity (Han et al., 2021; Zhao & Han, 2021), feature representations (Zhong et al., 2021a;b), or clustering methods (Fini et al., 2021; Zhang et al., 2023).

Recently, Vaze et al. (2022a) extended Novel Class Discovery into a more realistic scenario where the unlabeled data come from both novel and known classes, known as Generalized Category Discovery (GCD). To tackle this problem, GCD (Vaze et al., 2022a) adopts semi-supervised contrastive learning on the pre-trained visual transformer (Dosovitskiy et al., 2020). Meanwhile, ORCA (Cao et al., 2022) proposes an uncertainty adaptive margin mechanism to reduce the bias caused by the different learning speeds on labeled data and unlabeled data. Later, most concurrent works (Sun & Li, 2022; Zhang et al., 2022; Pu et al., 2023) focus on designing a better contrastive learning strategy to cluster novel classes. For example, PromptCAL (Zhang et al., 2022) uses auxiliary visual prompts in a two-stage contrastive affinity learning way to discover more reliable positive pairwise samples and perform more reasonable contrastive learning. DCCL (Pu et al., 2023) proposes a dynamic conceptional contrastive learning framework to alternately explore latent conceptional relationships between known classes and novel classes, and perform conceptional contrastive learning. However, those methods typically rely on transferring knowledge implicitly by sharing encoders, which can be restrictive as shown in (Gu et al., 2023).

In contrast, our method focuses on extracting and transferring known-class knowledge explicitly. Concurrently, Gu et al. (2023) distill knowledge in the model's output space in the standard NCD setting, but their method cannot be applied to the GCD setting directly due to its special design of weight function. In addition, they only consider a limited form of knowledge transfer while our proposed framework distills knowledge from the entire representation space of a pre-trained model of known classes, which is more flexible and effective.

**Knowledge Distillation** The concept of knowledge distillation was originally introduced by Hinton et al. Hinton et al. (2015), with the goal of transferring the "dark knowledge" from large models into small models. Based on the level of distilling, subsequent works can be categorized into two groups: distilling from logits and distilling from intermediate features. The first group (Zhao et al., 2022; Mirzadeh et al., 2020) which distills knowledge from logits primarily focuses on designing more effective knowledge distillation loss and optimization methods. The second group (Tian et al., 2020; Romero et al., 2014; Zagoruyko & Komodakis, 2016) considers that the intermediate features have richer knowledge and mainly focuses on directly transferring the feature representation or the similarity between samples. We refer the readers to Wang & Yoon (2021) for a more comprehensive survey on this topic.

In contrast to traditional knowledge distillation methods, which typically transfer knowledge within the same task, our approach transfers knowledge between two distinct tasks. Specifically, we extract knowledge from a model trained on known classes to guide the training of a model on both known classes and novel classes.

## 3 METHOD

### 3.1 PRELIMINARILY

We first introduce the setting of generalized category discovery problem (Vaze et al., 2022a), which aims to leverage known-class knowledge to discover novel classes. The dataset is composed of a labeled known classes set $\mathcal{D}^l = \{x_i^l, y_i^l\}_{i=0}^{|\mathcal{D}^l|}$ and an unlabeled set $\mathcal{D}^u = \{x_j^u\}_{j=0}^{|\mathcal{D}^u|}$, which contains both known and novel classes. Here $x, y$ represents the input image data and the corresponding label. In addition, we denote the number of known and novel classes as $C^k$ and $C^n$, and assume $C^n$ is known (Vaze et al., 2022a; Zhang et al., 2022). The goal is to classify known classes and cluster novel classes in $\mathcal{D}^u$ by leveraging $\mathcal{D}^l$.

Among almost all existing GCD methods (Vaze et al., 2022a; Wen et al., 2022; Zhang et al., 2022; Sun & Li, 2022), the basis of these models can be succinctly deconstructed into two components. The first component is the supervised learning of the labeled known class data. The second component is the unsupervised learning of both known and novel class unlabeled data. Therefore, the core part of their final loss function can be written as:

$$\mathcal{L}_{base} = (1 - \alpha)\mathcal{L}_s + \alpha\mathcal{L}_u, \tag{1}$$

where $\mathcal{L}_s$ is the supervised learning loss on labeled data, and $\mathcal{L}_u$ is the unsupervised learning loss on unlabeled data. $\alpha$ is a hyperparameter to balance the learning of labeled and unlabeled data.

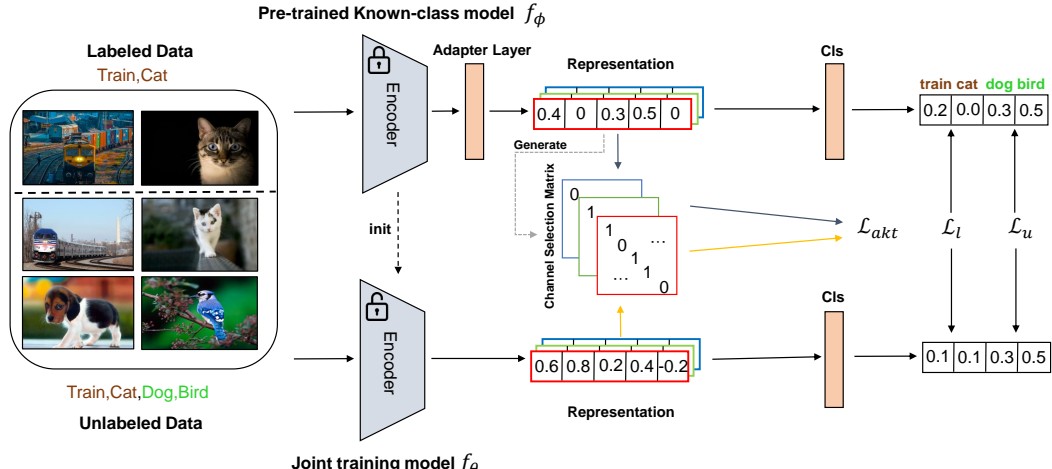

Figure 2: The overview of our adaptive knowledge transfer framework. Train, Cat and Dog, Bird represent known and novel classes. Cls denotes classifier. We discover novel classes by joint learning on all data and under the guidance of the pre-trained known-class model. To make our knowledge transfer framework more effective, we apply an adapter layer to transform features of the pre-trained known-class model, and utilize the transformed feature to generate an instance-wise Channel Selection Matrix to select channels to transfer. Finally, we maximize mutual information between two representation spaces in the selected channels by $\mathcal{L}_{akt}$.

### 3.2 MOTIVATIONS AND METHOD OVERVIEW

One of the main challenges in GCD is to transfer knowledge from known classes to novel classes. To achieve it, most existing methods (Vaze et al., 2022a; Wen et al., 2022; Zhang et al., 2022; Sun & Li, 2022) focused on the formulation of the unsupervised loss term $\mathcal{L}_u$ and establish a shared representation space for knowledge transfer from known to novel classes. However, this knowledge transfer is implicit which means it is hard to control the information flow between labeled and unlabeled data. Furthermore, our own experiment in Tab.1 have shown the inefficiency of this implicit knowledge transfer strategy in certain GCD problems.

To tackle this challenge, as shown in Fig.2, we propose an explicit knowledge transfer framework, comprising three core steps: 1) *Knowledge Generation*. This step is dedicated to generating the knowledge representation by a pre-trained known-class model $f_\phi$. 2) *Knowledge Alignment*. This step is specifically designed to enhance the transferability of the generated knowledge representation. 3) *Knowledge Distillation*. This step achieves knowledge transfer by maximizing the mutual information between two representation spaces. In the subsequent sections, we will provide a comprehensive explanation of each component in our novel adaptive knowledge transfer framework.

### 3.3 ADAPTIVE KNOWLEDGE TRANSFER

As discussed above, to transfer knowledge explicitly and effectively, we propose a novel adaptive knowledge transfer framework compromising three key components. In this section, we introduce the details of each component in turn.

**Knowledge Generation.** As demonstrated by our experiment in Tab.1, a pre-trained known-class model can effectively capture meaningful semantic clusters for novel classes, indicating that the pre-trained known-class model contains rich information useful for learning novel classes. In light of these observations, we posit that the pre-trained known-class model $f_\phi$ serves as an effective means to represent the knowledge pertaining to learning both the known and novel classes.

**Knowledge Alignment.** Although the representation space initialized with known classes contains rich information about the known-class data, it has not encountered novel classes during training. Consequently, this knowledge representation space is not well aligned with the all-class knowledge representation space. This misalignment can lead to ineffective knowledge transfer and bias the final

knowledge space towards known classes. To mitigate this issue, we propose a simple adapter layer $f_w$ that transforms this original representation space into the joint representation space. Specifically, our adapter layer consists of a linear and a ReLU layer. The original representation space is linearly transformed and truncated so that the transformed space can retain most of the original structure. We provide a detailed analysis of our adapter layer in the ablation study (Sec.4.3).

While we obtain the representation of known class knowledge, it is important to acknowledge that not all aspects of this knowledge are applicable to novel class learning. To select specific channels for knowledge transfer, we introduce a channel selection matrix. Since our adapter layer contains a ReLU layer, the channels whose values are less than 0 do not contribute to the final classification and clustering. Based on that, we assume that the channels whose values are greater than 0 are valuable for novel class learning. Therefore, we only transfer knowledge in the channels whose values are greater than 0. Consequently, we design our data-specific channel selection matrix $\mathbf{W}_s$ as:

$$\mathbf{W}_s = \mathrm{diag}(\mathbb{1}(\mathbf{v} > 0)), \tag{2}$$

where $\mathbb{1}$ is the indicator function, and $\mathbf{v} = f_w(f_\phi(\mathbf{x}))$. It is worth noting that $\mathbf{W}_s$ is a function of $\mathbf{v}$.

**Knowledge Distillation.** We implement our knowledge transfer by maximizing the mutual information between two representation spaces in the selected channels. In detail, let $\mathbf{U}$ and $\mathbf{V}$ represent the random variables corresponding to the representations of data in the joint model $f_\theta$ and the composite model of adapter layer and the pre-trained known-class model, respectively. Our objective is to maximize the mutual information between these two random variables in the selected channels,

$$\max_{\mathbf{U}} I(\mathbf{W}_s\mathbf{U}; \mathbf{W}_s\mathbf{V}), \tag{3}$$

where $\mathbf{U} = f_\theta(\mathbf{X})$, $\mathbf{V} = f_w(f_\phi(\mathbf{X}))$ and $\mathbf{X}$ is the random variable of the unlabeled data.

Specifically, inspired by InfoNCE proposed by Oord et al. (2018), we propose a contrastive representation loss, which utilizes noise-contrastive estimation to approximate mutual information in Equ.(3). See Appendix A for detail derivation. Specifically, we take the two representations of the same unlabeled data $x_i$ in two representation spaces, $\mathbf{u}_i = f_\theta(x_i)$, $\mathbf{v}_i = f_w(f_\phi(\mathbf{x}_i))$ as a positive pair meanwhile we take $\mathbf{u}_i$ and the generated features from the negative sample generator as negative pairs. Therefore, the adaptive knowledge transfer constraint term is formulated as:

$$\mathcal{L}_{akt} = -\frac{1}{|\mathcal{D}^u|} \sum_{i=1}^{|\mathcal{D}^u|} \log \frac{e^{\mathbf{u}_i^\top \mathbf{W}_s \mathbf{v}_i / \tau}}{e^{\mathbf{u}_i^\top \mathbf{W}_s \mathbf{v}_i / \tau} + \sum_{\mathbf{z} \in \mathcal{N}} e^{\mathbf{u}_i^\top \mathbf{W}_s \mathbf{z} / \tau}}, \tag{4}$$

where $\tau$ is the hyperparameter of temperature and $\mathcal{N}$ is the set of the generated negative samples in memory. In addition, we adopt a negative sample generation strategy based on a mixup scheme and provide a detailed implementation of $\mathcal{N}$ in Appendix E.

### 3.4 Learning Strategy

We adopt a two-stage learning strategy to learn our adaptive knowledge transfer framework. The first stage involves training the model $f_\phi$ on labeled known class data using the standard cross-entropy loss to extract the rough knowledge representation of known classes. In the second stage, to align representation, we first learn an adapter layer to transform the knowledge representation into joint representation space, and then utilize it to capture the channel selection matrix $\mathbf{W}_s$. To learn the adapter layer, we fix $f_\phi$, utilize the features of all data after the adapter layer to perform classification and clustering, and adopt $\mathcal{L}_s, \mathcal{L}_u$ to learn labeled and unlabeled data, respectively. The total loss is denoted as $(1 - \alpha)\mathcal{L}_s + \alpha\mathcal{L}_u$, which is the same as Equ.(1). To learn the joint representation space ($f_\theta$) and cosine classifier ($h$), we introduce a comprehensive loss function consisting of two components. The first component is the basis loss $\mathcal{L}_{base}$ of the labeled and unlabeled data learning. The second component is our proposed adaptive knowledge transfer loss $\mathcal{L}_{akt}$. In summary, the loss function of the joint representation space can be written as:

$$\mathcal{L} = \mathcal{L}_{base} + \beta\mathcal{L}_{akt} = (1 - \alpha)\mathcal{L}_s + \alpha\mathcal{L}_u + \beta\mathcal{L}_{akt} \tag{5}$$

where $\alpha, \beta$ is a hyperparameter that controls the weight of loss. In our final model, we set $\mathcal{L}_s$ as the typical cross-entropy loss on labeled data and $\mathcal{L}_u$ as the self-labeling loss on unlabeled data (Caron et al., 2021; Xu et al., 2022). We detail the $\mathcal{L}_u$ in Appendix D. It is worth noting that our method

does not depend on the design of $\mathcal{L}_u$. Furthermore, we empirically demonstrate the versatility of our approach by showing that $\mathcal{L}_{akt}$ yields performance enhancements across a spectrum of diverse designs for $\mathcal{L}_u$, as showcased in Tab.8.

In summary, our novel adaptive knowledge transfer loss enables the construction of an explicit knowledge transfer bridge between the labeled and unlabeled data. Consequently, the joint representation learning process retains the knowledge of known classes while also preserving the potential relationships between known and novel classes within the pre-trained known-class model.

## 4 EXPERIMENTS

### 4.1 EXPERIMENTAL SETUP

**Datasets.** To validate the effectiveness of our method, we conduct experiments on various datasets, including generic datasets such as CIFAR100 (Krizhevsky et al., 2009) and ImageNet100 (Deng et al., 2009), as well as the Semantic Shift Benchmark (Vaze et al., 2022b), namely CUB (Wah et al., 2011), Stanford Cars (Scars) (Krause et al., 2013), FGVC-Aircraft (Maji et al., 2013). In addition, as demonstrated by (Li et al., 2023), the semantic relationship between known and novel classes has a significant impact on novel class discovery. Therefore, to better evaluate the generalization capability of different methods, we propose a new benchmark dataset containing four splits with varying levels of semantic similarity between known and novel classes. Specifically, we select 20 classes from each of the 11 superclasses in the iNat21 dataset (Van Horn et al., 2021), totaling 220 classes. To create fine-grained splits, we divide each superclass into two halves, one for known classes and the other for unknown classes. For the other three splits, we utilize the CLIP model (Radford et al., 2021) to convert the superclass names into embeddings. We then randomly group these 11 superclasses into two groups and calculate the similarity between these two groups. A higher similarity between the two groups indicates a greater semantic similarity between the classes in these two splits. We rank these similarities and select different proportions to create Easy, Medium, and Hard splits. For more details, please refer to Appendix B.

Finally, for each dataset, we split the datasets into sets of known and novel classes. We take half of the data within the known classes as the labeled set $\mathcal{D}^l$, leaving the remaining portion unlabeled. As a result, $\mathcal{D}^u$ consists of unlabeled data from known classes and data from novel classes. The split details are in the Appendix C.

**Evaluation protocol.** Similar to (Vaze et al., 2022a), we evaluate the model on unlabeled datasets with clustering accuracy. Specifically, we first employ the Hungarian matching algorithm to obtain the best matching between cluster and ground truth, and then we report the performance separately on the known classes, novel classes, and all classes.

**Implementation details.** We adopt the DINO (Caron et al., 2021) pre-trained ViT-B/16 (Dosovitskiy et al., 2020) as our backbone, and we only finetune the last block of ViT-B/16. The adapter layer is composed of a linear and ReLU layer. In the first stage, we train our model by 20 epochs on labeled data. In the second stage, we train our model by 100 epochs on all data. We adopt the SGD optimizer with a momentum of 0.9, a weight decay of $5 \times 10^{-5}$, and an initial learning rate of 0.1, which reduces to $1e-4$ at 100 epoch using a cosine annealing schedule. The batch size is 128. The data augmentation is the same as (Vaze et al., 2022a). For hyperparameter, we follow (Vaze et al., 2022a; Xu et al., 2022) to set $\alpha = 0.35, \epsilon = 1$. Moreover, we follow (Caron et al., 2021) to set $\tau$ to 0.1, and $\tau'$ is initialized to 0.07, then warmed up to 0.04 with a cosine schedule in the starting 30 epochs. For the hyperparameter $\beta$ that we introduced, we set it to 0.1 for all datasets. We then validate its sensitivity in the ablation study. All the experiments are conducted on a single NVIDIA TITAN RTX.

### 4.2 COMPARISON WITH STATE-OF-THE-ART METHODS

Tab.2 presents the comparison results of our model with current state-of-the-art methods. Our results significantly outperform SOTA on almost all datasets. Specifically, on CIFAR100-80, our method achieves 2.3% improvement on novel classes. On ImageNet100-50, our method obtains 1.4% gains on novel classes. On three fine-grained datasets, our method outperforms the previous SOTA method on the overall metric by at least 3%. While on novel classes, we only drop 1% compared to DCCL (Pu et al., 2023) on CUB, and achieve 5.3% and 1.3% gains on Scars and Aircraft respectively.

Table 2: Comparison with state-of-the-art methods. We have adapted the original crNCD method for use in the GCD settings, and we refer to this adapted version as "crNCD*".

| Method | CIFAR100-80 | | | ImageNet100-50 | | | CUB | | | Scars | | | Aircraft | | |
|---|---|---|---|---|---|---|---|---|---|---|---|---|---|---|---|
| | All | Known | Novel | All | Known | Novel | All | Known | Novel | All | Known | Novel | All | Known | Novel |
| K-means | 52.0 | 52.2 | 50.8 | 72.7 | 75.5 | 71.3 | 34.3 | 38.9 | 32.1 | 12.8 | 10.6 | 13.8 | 16.0 | 14.4 | 16.8 |
| RS+ | 58.2 | 77.6 | 19.3 | 37.1 | 61.1 | 24.8 | 33.3 | 51.6 | 24.2 | 28.3 | 61.8 | 12.1 | 26.9 | 36.4 | 22.2 |
| UNO | 69.5 | 80.6 | 47.2 | 70.3 | 95.0 | 57.9 | 35.1 | 49.0 | 28.1 | 35.5 | 70.5 | 18.6 | 40.3 | 56.4 | 32.2 |
| ORCA | 69.0 | 77.4 | 52.0 | 73.5 | 92.6 | 63.9 | 35.3 | 45.6 | 30.2 | 23.5 | 50.1 | 10.7 | 22.0 | 31.8 | 17.1 |
| GCD | 70.8 | 77.6 | 57.0 | 74.1 | 89.8 | 66.3 | 51.3 | 56.6 | 48.7 | 39.0 | 57.6 | 29.9 | 45.0 | 41.1 | 46.9 |
| PromptCAL | 81.2 | 84.2 | 75.3 | 83.1 | 92.7 | 78.3 | 62.9 | 64.4 | 62.1 | 50.2 | 70.1 | 40.6 | 52.2 | 52.2 | 52.3 |
| DCCL | 75.3 | 76.8 | 70.2 | 80.5 | 90.5 | 76.2 | 63.5 | 60.8 | **64.9** | 43.1 | 55.7 | 36.2 | - | - | - |
| SimGCD | 78.1 | 77.6 | 78.0 | 82.4 | 90.7 | 78.3 | 60.3 | 65.6 | 57.7 | 46.8 | 64.9 | 38.0 | 48.8 | 51.0 | 47.8 |
| crNCD* | 80.4 | **85.3** | 70.6 | 81.7 | 91.3 | 76.9 | 64.1 | **75.2** | 58.6 | 54.8 | 76.5 | 44.3 | 53.1 | 57.0 | 51.3 |
| Ours | **82.8** | 84.0 | **80.3** | **84.1** | 92.8 | **79.7** | **67.1** | 73.7 | 63.8 | **59.2** | 79.1 | **49.6** | **55.9** | 60.7 | **53.6** |

Table 3: Results on proposed iNat21 benchmark

| Method | Fine-grain | | | Easy | | | Medium | | | Hard | | |
|---|---|---|---|---|---|---|---|---|---|---|---|---|
| | All | Known | Novel | All | Known | Novel | All | Known | Novel | All | Known | Novel |
| RS+ | 35.4 | 55.6 | 25.3 | 36.9 | 63.1 | 21.2 | 38.5 | 68.2 | 20.3 | 41.2 | 71.7 | 22.9 |
| UNO | 36.5 | 65.6 | 22.0 | 33.7 | 60.1 | 17.9 | 33.2 | 60.2 | 16.9 | 32.7 | 61.2 | 15.7 |
| GCD | 48.0 | 61.4 | 41.4 | 51.1 | 67.5 | 41.2 | 49.7 | 69.7 | 37.8 | 50.4 | 72.8 | 36.9 |
| SimGCD | 52.5 | 60.7 | 48.4 | 52.6 | 64.3 | 45.6 | 49.7 | 67.6 | 38.9 | 49.0 | 70.5 | 36.2 |
| crNCD* | 50.8 | 63.5 | 44.4 | 51.4 | 66.9 | 42.2 | 50.7 | 68.5 | 40.0 | 49.2 | 73.3 | 34.7 |
| Ours | **57.7** | **71.3** | **51.0** | **57.8** | **75.3** | **47.4** | **56.8** | **76.6** | **44.9** | **55.3** | **78.8** | **41.1** |

In Tab.3, we present the results on our proposed iNat21 benchmark. Our method demonstrates significant improvements over the previous state-of-the-art across all splits. Specifically, our approach achieves at least a 5% improvement on known classes while delivering more than a 2% gain on novel classes. Furthermore, when compared to SimGCD (Xu et al., 2022), our improvements are more pronounced on the medium and hard splits than on the easy and fine-grain splits. This observation suggests that knowledge transfer is more effective on splits with larger semantic differences. Additionally, we observe a trade-off between the performance of known classes and novel classes. Known classes perform best on the hard split but worst on the novel classes, whereas on the fine-grain split, known classes perform poorly while novel classes excel. In conclusion, the above results confirm the effectiveness of our proposed approach.

## 4.3 ABLATION STUDY

In this section, we conduct several analyses to evaluate and understand the effectiveness of our proposed method. We start with an ablation study to examine the contribution of each component in our approach. Next, we assess the representation ability of our pre-trained model on known classes to support our motivation and investigate the impact of different knowledge transfer designs. To gain visual insights, we visualize the representation space of different models, including the pre-trained model, adapter layer, and our proposed model. Additionally, we test our method in more challenging scenarios with limited labeling and fewer classes to assess its robustness. Meanwhile, we also present our model in a more realistic situation where the number of clusters is unknown. Finally, we analyze our approach performance with different $\mathcal{L}_u$. These analyses provide a comprehensive understanding of our approach and its effectiveness in various scenarios.

**Baseline and Cluster.** In Table 1, the baseline (BL) is trained by $\mathcal{L}_{base}$, which achieves competing results. And "Cluster" (Clu.) means we freeze the backbone pretrained by known classes and only learn cluster head for the unlabeled data by $\mathcal{L}_{base}$.

**Component analysis.** In Tab.4, we present the results of an ablation study to evaluate the effectiveness of three components in our model: naive Mutual Information loss (nMI), Adapter Layer (AL), and Channel Selection Matrix(CSM). Here, "nMI" means that we directly maximize the mutual information between the pre-trained known-class representation space and the joint representation space without AL and CSM. The results demonstrate that incorporating nMI significantly improves the performance of both novel and known classes across all datasets, highlighting the importance of explicit knowledge transfer. Furthermore, including AL results in a 1.3%, 2.7%, and -0.4% increase in all-class accuracy on fine-grained datasets. This indicates that AL effectively transforms the known-

Table 4: Ablation study. nMI, AL, and CSM denote naive mutual information loss, adapter layer, and channel selection matrix respectively.

| nMI | AL | CSM | CUB | | | Aircraft | | | Scars | | |
|-----|-----|-----|-----|-------|-------|-----|-------|-------|-----|-------|-------|
| | | | All | Known | Novel | All | Known | Novel | All | Known | Novel |
| | | | 61.7 | 68.0 | 58.5 | 49.6 | 56.3 | 46.2 | 51.8 | 71.9 | 42.0 |
| ✓ | | | 65.5 | 71.9 | 62.2 | 52.9 | 59.5 | 49.6 | 58.0 | 77.4 | 48.5 |
| ✓ | ✓ | | 66.8 | **75.6** | 62.5 | 55.6 | 60.5 | 53.1 | 57.6 | 75.9 | 48.8 |
| ✓ | ✓ | ✓ | **67.1** | 73.7 | **63.8** | **55.9** | **60.7** | **53.6** | **59.2** | **79.1** | **49.6** |

Table 5: Adapter layer designs.

| Layer | 0 | 1 | 2 |
|-------|------|------|------|
| AL | 62.7 | 56.9 | 48.2 |
| Ours | 62.2 | 63.8 | 58.0 |

Table 6: Comparison of different knowledge transfer designs.

| Method | CUB | | | Aircraft | | | Scars | | |
|--------|-----|-------|-------|-----|-------|-------|-----|-------|-------|
| | All | Known | Novel | All | Known | Novel | All | Known | Novel |
| Baseline | 61.7 | 68.0 | 58.5 | 49.6 | 56.3 | 46.2 | 51.8 | 71.9 | 42.0 |
| +MSE | 63.8 | 70.9 | 60.2 | 51.8 | 57.6 | 48.9 | 54.1 | 76.8 | 43.2 |
| +KL | 63.7 | **74.2** | 58.4 | 51.0 | 55.3 | 48.9 | 55.7 | **77.6** | 45.1 |
| +nMI | **65.5** | 71.9 | **62.2** | **52.9** | **59.5** | **49.6** | **58.0** | 77.4 | **48.5** |

class representation into a more suitable form for novel classes, enhancing the transfer efficiency. Finally, adding CSM to our model leads to improvements of 1.3%, 0.5%, and 0.8% on novel classes for CUB, Aircraft, and Scars datasets, respectively. Simultaneously, the performance of known classes on the three fine-grained datasets increases by -1.9%, 0.2%, and 3.2%. This outcome aligns with our original design intention that using CSM to select the relevant channels for novel classes to enhance knowledge transfer efficiency. Additionally, we offer a visual analysis of CSM in Appendix K. Overall, the results demonstrate each component's effectiveness in our model, highlighting their contributions to improving the performance of known and novel classes.

**Variant knowledge transfer designs.** Tab.6 provides an analysis of different implementations of the knowledge transfer without the knowledge alignment. Here "KL" denotes the knowledge distillation loss term proposed in (Gu et al., 2023) but without their additional weight function design. First, we observe that even with the simplest mean squared error (MSE) loss, the model achieves significant performance improvements across all datasets. Specifically, with MSE, the model shows improvements of 2.1%, 2.2%, and 2.3% in all class accuracy on CUB, Aircraft, and Scars datasets, respectively, demonstrating the importance of explicit knowledge transfer. Furthermore, comparing the results of nMI with KL, we observe a significant improvement of 1-2% on all datasets with nMI which indicates the superiority of our method.

**Adapter layer analysis.** In this part, we analyze the design of our adapter layer. In Fig.3, we visualize representation spaces before and after the adapter layer by t-SNE. After passing through the adapter layer, the representation space of the model undergoes a notable transformation from a disorganized and scattered representation found in the pre-trained known-class model to a more compact and structured configuration. This observation is in line with the intended purpose of our adapter layer design, which aims to transform the representation space of the pre-trained known-class model into the joint space. Furthermore, it is worth noting that compared to the adapter layer, our final model can exhibit more compact feature representations, with samples of the same class tightly grouped together, further confirming our method's effectiveness. Additionally, in Tab.5, we compare the model's performance on CUB novel classes with different numbers of layers, both with and without knowledge transfer. Each layer in our adapter layer consists of a linear layer and a ReLU layer. Upon analyzing the results after the adapter layer, we observe that the model with 0 layers (i.e., without additional layers) outperforms the models with more layers. This finding can be attributed to the noise generated during unsupervised learning. More complex models are susceptible to overfitting this noise, leading to a decrease in model generalization performance. Our analysis concludes that the one-layer design is the optimal choice for our proposed adaptive knowledge transfer. This design effectively converts the known-class representation space into a joint representation space.

**The number of clusters $C^n$ is unknown.** The experiments presented so far have relied on the assumption that the number of clusters is known a priori, which is often unrealistic in practice. To address this limitation, we conduct additional experiments to analyze our model's performance when the number of clusters is unknown. In this experiment, we employ the method proposed in (Vaze et al., 2022a) to infer the number of classes for each dataset. Specifically, we consider Aircraft to have 108 classes, CUB to have 231 classes, and Scars to have 230 classes. The results presented in Tab.7 show that our method significantly improves over the baseline for known and novel classes.

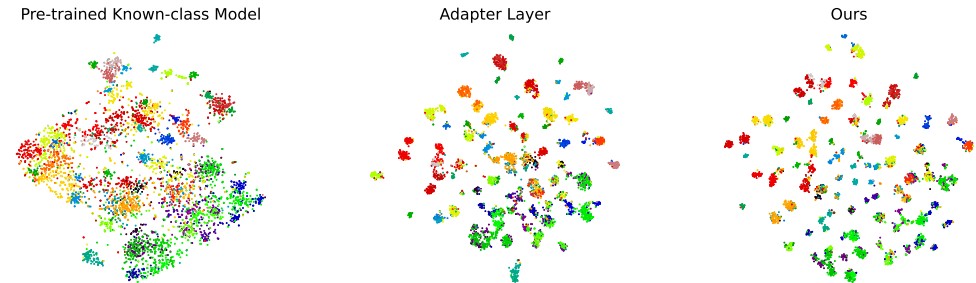

Figure 3: t-SNE visualization of data features on Aircraft. Here "Pre-trained Known-class Model" is the pre-trained model representation space before the Adapter Layer. The middle figure is the pre-trained model representation space after the Adapter Layer. Moreover, we also present our final model representation space with the label "Ours".

Table 7: $C^n$ is unknown.

| Method | CUB | | | Scars | | | Aircraft | | |
|---|---|---|---|---|---|---|---|---|---|
| | All | Known | Novel | All | Known | Novel | All | Known | Novel |
| Baseline | 62.4 | 67.1 | 60.0 | 52.6 | 72.7 | 42.9 | 52.3 | 56.2 | 50.3 |
| Ours | 68.0 | 72.5 | 65.8 | 54.9 | 75.3 | 45.1 | 55.6 | 60.4 | 53.1 |

Table 8: Different $\mathcal{L}_u$ loss.

| Method | CUB | | | Scars | | | Aircraft | | |
|---|---|---|---|---|---|---|---|---|---|
| | All | Known | Novel | All | Known | Novel | All | Known | Novel |
| $\mathcal{L}_u^{pw}$ | 33.2 | 24.0 | 37.8 | 34.2 | 31.3 | 35.7 | 17.2 | 18.3 | 16.7 |
| $\mathcal{L}_u^{pw} + \mathcal{L}_{akt}$ | 38.6 | 36.6 | 39.5 | 35.1 | 25.6 | 39.9 | 26.8 | 39.4 | 20.8 |
| $\mathcal{L}_u^{ot}$ | 53.9 | 63.0 | 49.3 | 42.3 | 47.5 | 39.7 | 40.4 | 45.6 | 37.9 |
| $\mathcal{L}_u^{ot} + \mathcal{L}_{akt}$ | 58.7 | 59.1 | 58.5 | 48.8 | 47.8 | 49.2 | 49.8 | 63.1 | 43.3 |

These findings underscore the robustness of our approach in a realistic setting. Notably, our model's performance on the CUB dataset under a wrong estimate is better than that under the correct estimate. This may be attributed to overestimating the number of novel classes reduces the occurrence of extremely large clusters. Further analysis is provided in Appendix J.

**Analysis of different $\mathcal{L}_u$.** In this part, we demonstrate that our knowledge transfer framework is not dependent on the design of $\mathcal{L}_u$. Specifically, we conduct experiments with two different unsupervised loss functions: the typical pairwise loss proposed by (Cao et al., 2022), denoted as $\mathcal{L}_u^{pw}$, and the optimal transport-based self-labeling loss proposed by (Fini et al., 2021), denoted as $\mathcal{L}_u^{ot}$. As shown in Tab.8, our $\mathcal{L}_{akt}$ loss significantly improves the performance of both novel and known classes. Notably, even with the relatively strong $\mathcal{L}_u^{ot}$, we achieve a substantial improvement in the performance of novel classes, highlighting the effectiveness of our method with different $\mathcal{L}_u$ formulations.

## 5 CONCLUSION

This paper introduces a novel adaptive knowledge transfer framework for generalized category discovery aiming at establishing an explicit knowledge transfer between known classes and novel classes. The framework comprises three essential components: knowledge generation, knowledge alignment, and knowledge distillation. In the knowledge generation component, we utilize a model trained on known class data. The knowledge alignment consists of two submodules: an adapter layer, which can align the known-class representation space with the joint representation space, and a channel selection matrix, which facilitates more targeted knowledge transfer. Finally, our knowledge distillation component focuses on maximizing the mutual information between two representation spaces. Our extensive evaluations reveal the remarkable superiority of our approach when compared to existing methods in the field. Moreover, our explicit knowledge transfer framework introduces a fresh perspective for advancing knowledge transfer in generalized category discovery. This novel approach holds promise for addressing the critical challenge of effectively transferring knowledge from known to novel classes in GCD.

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

# A ESTIMATING OF MUTUAL INFORMATION

For notation simplicity, we derive the mutual information between $X$ and $Y$ in the following. The mutual information between $X$ and $Y$ is defined as:

$$I(X;Y) = \mathbb{E}_{p(x,y)}[\log \frac{p(x,y)}{p(x)p(y)}] \tag{6}$$

However, the above equation is intractable due to $p(x), p(y)$ is unknown, we turn to model the density ratios:

$$f(x,y) \propto \frac{p(x,y)}{p(x)p(y)} \tag{7}$$

We will prove this equation later. Note that the density ratio can be modeled by a neural network $f_\theta(x,y)$, where $\theta$ is the parameters of the neural network. We simply use the following form:

$$f_\theta(x,y) = \exp(xg_\theta(z)) \tag{8}$$

Although we cannot evaluate $p(y)$ or $p(y|x)$ directly, we can sample from these distributions, and then use the Noise-Contrastive Estimation to approximate mutual information. Specifically, given a set $U = \{y_1, y_2, ..., y_N\}$, N - 1 negative samples from the proposal distributions $p(y)$ and one positive sample from $p(y|x)$ denoted as $y_N$, we minimize the following objective:

$$\mathcal{L} = \mathbb{E}_U[\log \frac{f_\theta(x, y_N)}{\sum_{i=1}^N [f_\theta(x, y_i)]}] \tag{9}$$

Now, we prove that minimizing $\mathcal{L}$ is equivalent to maximizing $I(X;Y)$. First, we show the $\mathcal{L}$ achieve optimal when $f(x,y) \propto \frac{p(x,y)}{p(x)p(y)}$. As $\mathcal{L}$ is the cross-entropy loss to classify positive samples correctly, and $N = i$ denotes $y_i$ is the positive sample from $p(y|x)$. We write the optimal probability as:

$$p(N = i|X, Y) = \frac{p(y_i|x)\prod_{j \neq i} p(y_j)}{\sum_{j=1}^N p(y_j|x)\prod_{k \neq j} p(y_k)}. \tag{10}$$

Dividing the numerator and denominator simultaneously by $\prod_{i=1}^N p(y_i)$, the above equation turns to the following form:

$$p(N = i|X, Y) = \frac{\frac{p(y_i|x)}{p(y_i)}}{\sum_{j=1}^N \frac{p(y_j|x)}{p(y_j)}}. \tag{11}$$

Therefore, the optimal value for $f_\theta(x, y_i)$ in $\mathcal{L}$ is proportional to $\frac{p(y_i|x)}{p(y_i)}$.

Then, we plug in the optimal $f_\theta(x, y_i)$ in $\mathcal{L}$. The optimal loss is denoted as:

$$\begin{aligned}
\mathcal{L}^{opt} &= -\mathbb{E}[\log \frac{\frac{p(y_N|x)}{p(y_N)}}{\sum_{j=1}^N \frac{p(y_j|x)}{p(y_j)}}] \\
&= \mathbb{E}[\log 1 + \frac{p(y_N)}{p(y_N|x)} \sum_{j=1}^{N-1} \frac{p(y_j|x)}{p(y_j)}] \\
&\approx \mathbb{E}[\log 1 + \frac{p(y_N)}{p(y_N|x)}(N-1)\mathbb{E}_{y \in p(y)} \frac{p(y_j|x)}{p(y_j)}] \\
&= \mathbb{E}[\log 1 + \frac{p(y_N)}{p(y_N|x)}(N-1)] \\
&\geq \mathbb{E}[\log \frac{p(y_N)}{p(y_N|x)}N] \\
&= -I(X;Y) + \log N.
\end{aligned}$$

Therefore, $I(X;Y) \geq \log N - \mathcal{L}^{opt} \geq \log N - \mathcal{L}$. In conclusion, we minimize $\mathcal{L}$ $w.r.t$ $\theta$ to approximate $I(X;Y)$. For more in-depth analysis and proof please see InfoNCE (Van den Oord et al., 2018).

## B  INAT21 BENCHMARK CONSTRUCTION

To manage computational costs effectively, we randomly selected 20 fine-grained classes from each of the 11 super categories contained within the iNat21 dataset. The detailed benchmark is further constructed by partitioning it into four distinct splits. These splits were strategically organized based on semantic similarity and a hierarchical, coarse-to-fine structure within the classes. This methodical division ensures that the proposed benchmark encapsulates a rich variety of class relationships and hierarchies, thereby facilitating a more comprehensive assessment of knowledge transfer techniques.

### B.1  FINE-GRAIN SPLITS

For fine-grain split, we simply sample 10 classes from each super-category. In this specific split, we've introduced a noteworthy characteristic: the unlabeled classes share coarse-level label information with their labeled counterparts. By incorporating this fine-grained setting, we aim to create a more realistic and demanding evaluation scenario for knowledge transfer techniques.

### B.2  CORASE SPLITS

**Similarity Computation.**  With the success of the Vision-Language Model such as CLIP (Radford et al., 2021), text and vision information are well aligned in the feature space. Leveraging the powerful representations learned by these models, we propose to use the similarity between class labels to approximate the semantic similarity of a large amount of visual data.
Specifically, given two sets of labels $Y_1 = [y_{1,1}, y_{1,2}, ...y_{1,N_1}]$, $Y_2 = [y_{2,1}, y_{2,2}, ...y_{2,N_2}]$ and a Vision-Language model $G$, class embeddings are computed by $f_1 = G(Y_1)$ and $f_2 = G(Y_2)$. Next, a distance matrix is calculated by considering cosine similarity between feature embeddings, i.e. $d = \text{norm}(f_1) \cdot \text{norm}(f_2^T)$.
We make the assumption that classes close to each other in the feature space contain valuable semantic information for the discovery of novel classes. Conversely, distant classes have limited relevance for this task. Thus, the final distance for the two label sets is calculated by the average of $k$ smallest measure for each label in $Y_1$, i.e.

$$D = \frac{1}{N_1 \cdot k} \sum_{i=1}^{N_1} \sum_{j=1}^{k} \hat{d}_{i,j} \tag{12}$$

where $\hat{d}_{i,j}$ denotes the $j^{th}$ smallest distance in $d_i$. This distance measure is used to represent the semantic similarity of two sets of data. This approach is able to capture the semantic relationships between labels in a data-driven manner, emphasizing the most informative and closely related labels while disregarding those that are distant and less relevant.

**Corase Split design.**  For our detailed splits, we consider the 11 super-category labels in the original iNat21 dataset (Van Horn et al., 2021), which 5 of them will be novel classes and the rest known. Specifically $N_1 = 5$, $N_2 = 6$ and we set $k = 1$, so for all the possible splits in the 11 superclasses, we can calculate a list of similarity measure $D_{i=1}^{k}$ where $k = \binom{11}{5}$. We then arrange all these similarity measures in descending order and strategically select 0%, 60%, and 100% of the sorted list to create hard, medium, and easy splits, respectively. This approach ensures that the 'hard' split exhibits the lowest similarity between known and novel classes, while the 'easy' split demonstrates the highest, and the 'medium' split achieves a balanced similarity level. Further, all the split we introduce creates a different scenario that both known/novel classes are fine-grained but do not share corase level information, which is more challenging for the knowledge transfer task.

## C  DATASETS

We conduct experiments on widely-used datasets such as CIFAR100 (Krizhevsky et al., 2009) and ImageNet100 (Deng et al., 2009), as well as the recently proposed Semantic Shift Benchmark (Vaze

et al., 2022b), namely CUB (Wah et al., 2011), Stanford Cars (Scars) (Krause et al., 2013) and FGVC-Aircraft (Maji et al., 2013). The details of the split are as follows:

Table 9: The detail of datasets.

| Dataset | Labeled $\mathcal{D}^l$ | | Unlabeled $\mathcal{D}^u$ | |
|---------|-------------------------|--------|---------------------------|--------|
| | #Image | #Class | #Image | #Class |
| CIFAR100 | 20K | 80 | 30k | 100 |
| ImageNet100 | 31.9K | 50 | 95.3K | 50 |
| CUB | 1.5K | 100 | 4.5K | 200 |
| Stanford Cars | 2.0K | 98 | 6.1K | 196 |
| FGVC-Aircraft | 1.7K | 50 | 5.0K | 100 |

## D  THE DETAILS OF $\mathcal{L}_u$

In this paper, we adopt the self-labeling loss  (Caron et al., 2021; Wen et al., 2022) as our $\mathcal{L}_u$. Specifically, for each unlabeled data point $x_i$, we generate two views $x_i^{v_1}$ and $x_i^{v_2}$ through random data augmentation.  These views are then fed into the ViT (Dosovitskiy et al., 2020) encoder and cosine classifier ($h$), resulting in two predictions $\mathbf{y}_i^{v_1} = h(f_\theta(x_i^{v_1}))$ and $\mathbf{y}_i^{v_2} = h(f_\theta(x_i^{v_2}))$, $\mathbf{y}_i^{v_1}, \mathbf{y}_i^{v_2} \in \mathbb{R}^{C^k + C^n}$. As we expect the model to produce consistent predictions for both views, we employ $\mathbf{y}_i^{v_2}$ to generate a pseudo label for supervising $\mathbf{y}_i^{v_1}$. The probability prediction and its pseudo label are denoted as:

$$\mathbf{p}_i^{v_1} = \texttt{Softmax}(\mathbf{y}_i^{v_1}/\tau), \quad \mathbf{q}_i^{v_2} = \texttt{Softmax}(\mathbf{y}_i^{v_2}/\tau') \tag{13}$$

Here, $\tau, \tau'$ represents the temperature coefficients that control the sharpness of the prediction and pseudo label, respectively. Similarly, we employ the generated pseudo-label $\mathbf{q}_i^{v_1}$, based on $\mathbf{y}_i^{v_1}$, to supervise $\mathbf{y}_i^{v_2}$. However, self-labeling approaches may result in a degenerate solution where all novel classes are clustered into a single class (Caron et al., 2018). To mitigate this issue, we introduce an additional constraint on cluster size. Thus, the loss function can be defined as follows:

$$\mathcal{L}_u = \frac{1}{2|\mathcal{D}^u|} \sum_{i=1}^{|\mathcal{D}^u|} [l(\mathbf{p}_i^{v_1}, \texttt{SG}(\mathbf{q}_i^{v_2})) + l(\mathbf{p}_i^{v_2}, \texttt{SG}(\mathbf{q}_i^{v_1}))] + \epsilon \mathbf{H}(\frac{1}{2|\mathcal{D}^u|} \sum_{i=1}^{|\mathcal{D}^u|} \mathbf{p}_i^{v_1} + \mathbf{p}_i^{v_2}) \tag{14}$$

Here, $l(\mathbf{p}, \mathbf{q}) = -\mathbf{q} \log \mathbf{p}$ represents the standard cross-entropy loss, and $\texttt{SG}$ denotes the "stop gradient" operation. The entropy regularizer $\mathbf{H}$ enforces cluster size to be uniform thus alleviating the degenerate solution issue. The parameter $\epsilon$ represents the weight of the regularizer.

## E  THE DETAILS OF $\mathcal{N}$

Due to the presence of samples from the same class in a simple negative samples set $\mathcal{N}$, minimizing $\mathcal{L}_{akt}$ may have detrimental effects on clustering (Zheng et al., 2021; Zhang et al., 2022; Zhong et al., 2021b), particularly for novel classes. To address this concern, we propose a negative sample generation strategy based on mixup. The use of contrastive loss poses a potential issue as it may mistakenly treat different unlabeled data samples from the same class as negative samples, which hinders the effective clustering of novel classes (Zhang et al., 2022; Zheng et al., 2021). To address this concern, we generate negative samples by combining the representation of labeled and unlabeled data. Specifically, we mix the representations of labeled and unlabeled data as follows:

$$\mathcal{N} = \{\mathbf{z} | \mathbf{z} = \eta \mathbf{z}^l + (1 - \eta) \mathbf{z}^u, \eta \in (0.5, 1]\} \tag{15}$$

Here, $\mathbf{z}^l$ and $\mathbf{z}^u$ represent the representations of labeled and unlabeled data in two representation spaces, and $\eta$ is a random value between 0.5 and 1. Because $\eta > 0.5$, the generated negative samples tend to be biased towards the known classes since the labeled data belong to known classes. Consequently, this approach helps to avoid class collision issues for novel classes. As the known classes already have supervised losses, the negative impact of the contrastive loss on the classification of known classes is relatively small. Furthermore, as shown in Tab.10, using NSG can slightly improve the model's performance, especially on novel classes.

Table 10: Comparison of model's performance with and without NSG.

| Method | CUB | | | StanfordCars | | | Aircraft | | |
|---|---|---|---|---|---|---|---|---|---|
| | All | Seen | Novel | All | Seen | Novel | All | Seen | Novel |
| Ours w/o NSG | 67.2 | 73.2 | 64.2 | 58.9 | 80.2 | 48.6 | 54.1 | 60.3 | 51.0 |
| Ours | 67.1 | 73.7 | 63.8 | 59.2 | 79.1 | 49.6 | 55.9 | 60.7 | 53.6 |

Table 11: Comparison with state-of-the-art Method

| Method | Herbarium | | |
|---|---|---|---|
| | All | Seen | Novel |
| K-means | 12.9 | 12.9 | 12.8 |
| Rankstats | 27.9 | 55.8 | 12.8 |
| UNO | 28.3 | 53.7 | 14.7 |
| ORCA | 20.9 | 30.9 | 15.5 |
| GCD | 35.4 | 51 | 27 |
| OpenCons | 39.3 | 58.9 | 28.6 |
| PromptCAL | - | - | - |
| DCCL | - | - | - |
| SimGCD | 43.3 | 57.9 | 35.3 |
| crNCD | - | - | - |
| Ours | 43.0 | 56.2 | 35.9 |

## F  RESULTS ON HERBARIUM19 DATASET.

We conduct the experiment on the Herbarium19 dataset, which is a long-tailed dataset. The results in Tab.11 show we achieve comparable results on the Herbarium19 with SimGCD, while PromptCAL, DCCL, and crNCD do not report results on the Herbarium19 dataset.

## G  MORE ABLATION ON COARSE-GRAINED DATASET

We conduct an ablation study on the coarse-grained dataset. The observation in the table below regarding the coarse-grained dataset reveals a significant drop in novel class performance after using nMI. We attribute this to two main reasons. a) The representation space initialized with known classes has not encountered novel classes during training, potentially introducing bias in the final knowledge space towards known classes. This motivated the design of AL in our approach. b) The knowledge in known class data is not universally useful, and some information may even be harmful to novel class learning. This phenomenon is more pronounced in coarse-grained datasets. Consequently, we introduce CSM to address this issue by filtering out harmful knowledge. We provide a detailed analysis of this phenomenon in the subsequent section.

The table results highlight the limitations of directly using nMI, particularly in scenarios with a large semantic gap between classes. The subsequent improvement seen after employing AL and CSM underscores their efficacy in enabling more targeted knowledge transfer for novel class learning.

## H  THE DETAILS OF CLUSTERING ON THE PRE-TRAINED MODEL

We initialize $C^k + C^n$ prototypes, and then use $(1 - \alpha)\mathcal{L}_s + \alpha\mathcal{L}_u$ to learn prototypes on all data. Finally, we evaluate the model on unlabeled datasets with clustering accuracy. Specifically, we first employ the Hungarian matching algorithm to obtain the best matching between cluster and ground truth, and then we report the performance on the known classes, novel classes, and all classes. More results are shown in Tab.13

## I  MORE IMPLEMENTATION DETAILS OF CRNCD

As crNCD is originally designed for NCD problems, we have adapted it to the GCD setting using a widely employed architecture utilized in other GCD methods. Additionally, to mitigate the influence

Table 12: Ablation study on coarse-grained dataset.

| nMI | AL | CSM | CIFAR100-80 | | |
|---|---|---|---|---|---|
| | | | All | Known | Novel |
| | | | 79.2 | 78.7 | 80.0 |
| ✓ | | | 79.9 | 84.6 | 70.5 |
| ✓ | ✓ | | 82.2 | 84.1 | 78.3 |
| ✓ | ✓ | ✓ | 82.8 | 84.0 | 80.3 |

Table 13: More analysis of the pre-trained known-class model. "Clustering" denotes clustering novel classes on the pre-trained model.

| Method | CIFAR100-80 | | | ImageNet100-50 | | | CUB | | | Scars | | | Aircraft | | |
|---|---|---|---|---|---|---|---|---|---|---|---|---|---|---|---|
| | All | Known | Novel | All | Known | Novel | All | Known | Novel | All | Known | Novel | All | Known | Novel |
| Baseline | 79.3 | 78.4 | 80.9 | 83.6 | 93.3 | 78.7 | 61.7 | 68.0 | 58.5 | 49.6 | 56.3 | 46.2 | 51.8 | 71.9 | 42.0 |
| Clustering | 57.6 | 81.6 | 45.6 | 80.1 | 96.7 | 71.7 | 67.3 | 76.5 | 62.7 | 52.5 | 75.2 | 41.5 | 48.0 | 59.0 | 42.5 |

of different self-labeling designs, we employ the same $\mathcal{L}_{base}$ (Equ.1) as in our method, instead of using the original $\mathcal{L}_{opt}$ from crNCD. The original $\mathcal{L}_{opt}$ in crNCD did not perform well in the GCD setting, as indicated in the table below. It is important to note that we have made no modifications to the core loss $\mathcal{L}_{rKT}$.

## J   MORE ANALYSIS ON $C^n$ IS UNKNOWN

In the experiment section, we have presented the performance of our model when the number of novel classes $C^n$ is unknown. Notably, we observed that the model achieves better performance in the $C^n$ unknown scenario compared to the $C^n$ known scenario on the CUB dataset. In this section, we aim to provide some insights into this phenomenon.

In our experiment setup, we select a subset of the CUB unlabeled training dataset, where the number of samples in a novel class ranges between 29 and 30, while the number of samples in a known class is around 15.

In Fig.4, we provide visual representations of the confusion matrices corresponding to the $C^n$ known and $C^n$ unknown scenarios. Notably, when $C^n$ is unknown, we observe that the model exhibits a low tendency to predict data belonging to the additional classes (ranging from 200 to 230). Furthermore, the confusion matrix appears to exhibit a more sparse structure in the $C^n$ unknown scenario. Specifically, when compared to the $C^n$ known scenario, the $C^n$ unknown scenario displays a decreased number of instances with large values in the confusion matrix. Specifically, in the $C^n$ known scenario, we observe five instances of large incorrect values exceeding 20, whereas, in the $C^n$ unknown scenario, only 3 such instances are present. This finding indicates that in the $C^n$ unknown scenario, the model is less inclined to generate erroneous sub-classes, resulting in a sparser confusion matrix.

In Fig.5, we present the histograms of cluster sizes specifically focusing on the novel classes. Interestingly, we observe that in the $C^n$ unknown scenario, there is a lower occurrence of large clusters with sizes ranging between 50 and 60. This observation further supports our findings that in the $C^n$ unknown scenario, the model is less prone to generating large clusters.

## K   CHANNEL SELECTION MATRIX VISUALIZATION ANALYSIS

To visualize the pattern of the Channel Selection Matrix, we first extract the Channel Selection Matrix for each sample. Next, we aggregate the Channel Selection Matrix for each class by summing them. Finally, we apply normalization to the aggregated values for each class. The diagonal of the aggregated Channel Selection Matrix is shown in Fig.6. Note that for fine-grained datasets, the classes often exhibit similarity when their indexes are close. The pattern of the Channel Selection Matrix is evident and distinct for the fine-grained dataset, particularly for the aircraft dataset. However, for ImageNet100, the pattern appears less clear. In Fig.6, we observe that for each class, specific dimensions are activated in the Channel Selection Matrix. Additionally, when classes are similar, the activations also exhibit similarity, indicating a correlation between class similarity and mask

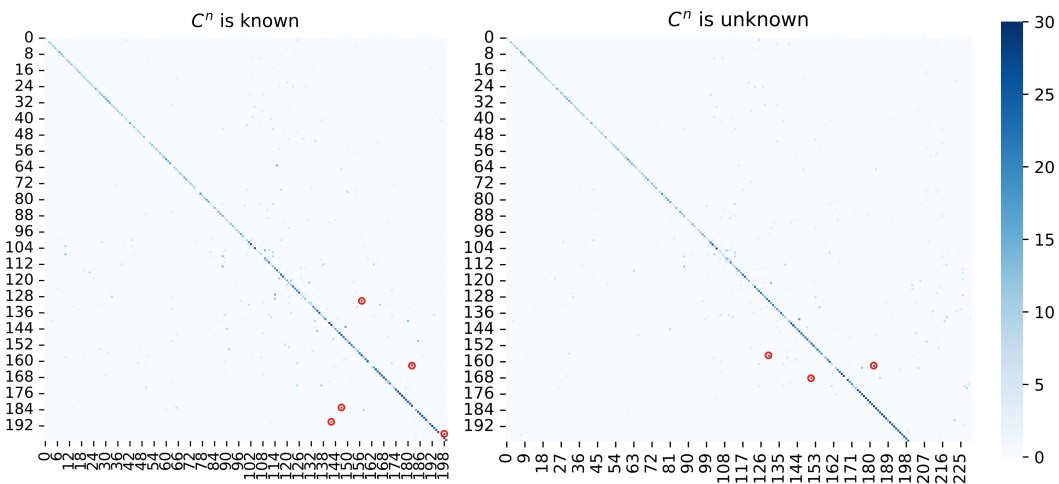

Figure 4: Comparisons of confusion matrix on different settings in CUB dataset. The x-axis is the predicted label, and the y-axis is the true label. Red circles mark values greater than 20 that are not on the diagonal.

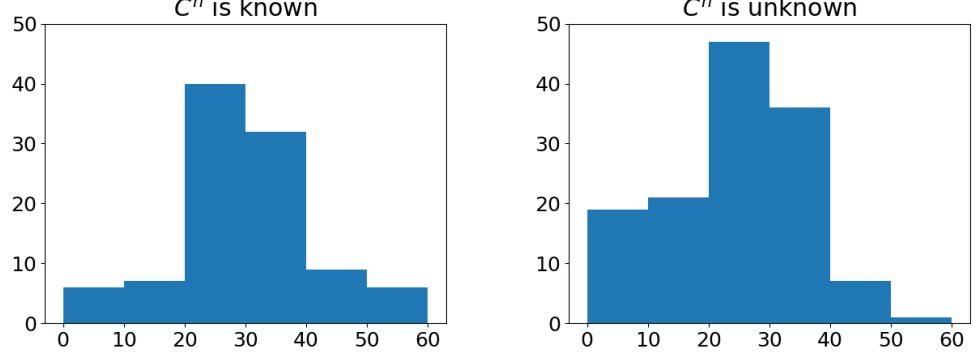

Figure 5: Comparisons of cluster size histograms on novel classes in CUB dataset. The true cluster size is between 29 and 30. Notably, in the $C^n$ unknown scenario, because of the additional class, there are 31 extra items in the $C^n$ unknown histogram.

Table 14: crNCD performance with original loss $\mathcal{L}_{opt}$

| Method | CUB | | | StanfordCars | | | Aircraft | | |
|---|---|---|---|---|---|---|---|---|---|
| | All | Seen | Novel | All | Seen | Novel | All | Seen | Novel |
| $\mathcal{L}^{opt}$ + crNCD | 56.9 | 57.8 | 56.4 | 46.6 | 62.3 | 39.1 | 45.2 | 59.0 | 43.4 |
| $\mathcal{L}_{base}$ + crNCD | 64.1 | 75.2 | 58.6 | 54.8 | 76.5 | 44.3 | 53.1 | 57.0 | 51.3 |

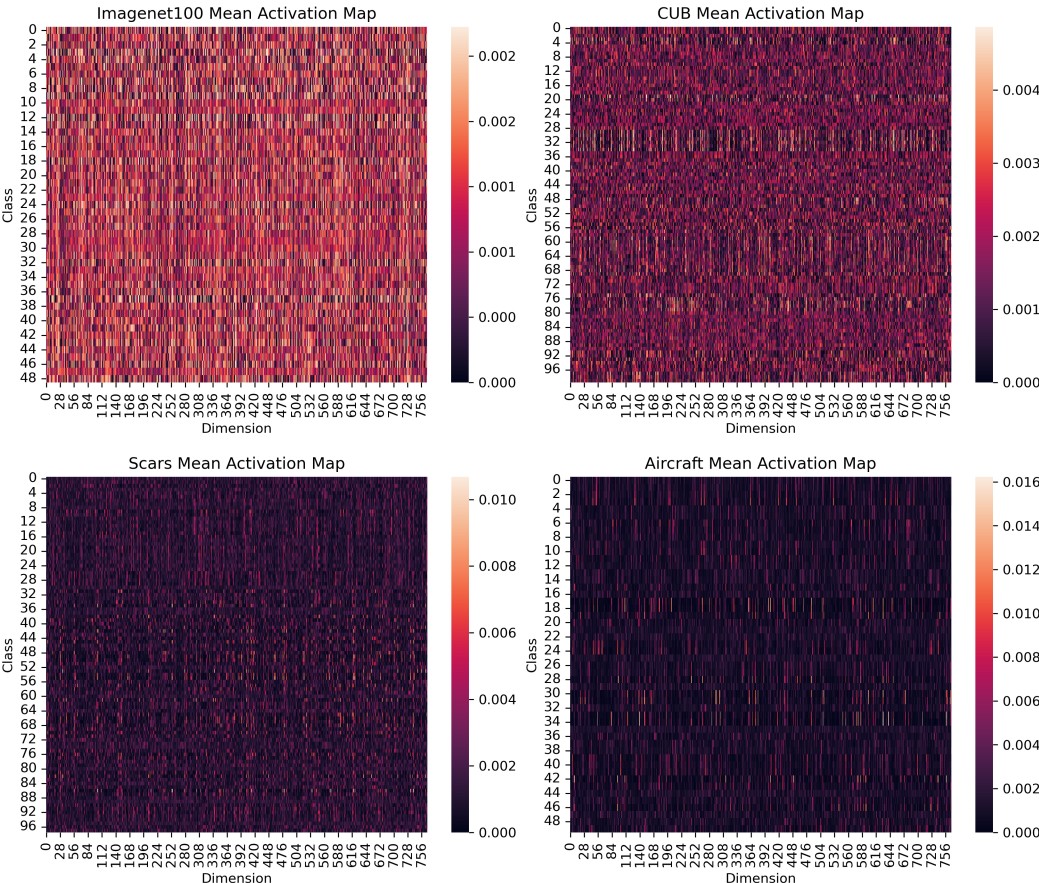

Figure 6: Visualization of the Channel Selection Matrix. The diagonal of the Channel Selection Matrix is shown.

activations. This observation aligns perfectly with our initial motivation behind the Channel Selection Matrix. Our approach selects different features to transfer for different classes, tailoring the knowledge transfer process according to the unique characteristics of each class. Meanwhile, the unselected features are unconstrained, enhancing the flexibility of the joint training model.

## L    FEWER KNOWN CLASSES AND LOWER LABELING RATIOS

In Fig.7, we evaluate our proposed method in two scenarios: (1) we reduce the number of known classes while annotating half of each known class, and (2) we reduce the labeling ratios while given 100 classes as known classes. The results demonstrate that our method surpasses the baseline model in both scenarios. In the fewer known classes situation, our method achieves a significant improvement of 3-7%. Similarly, in the lower labeling ratios situation, we observe a 1-9% improvement. These results highlight the effectiveness and robustness of our method in addressing more challenging and realistic scenarios.

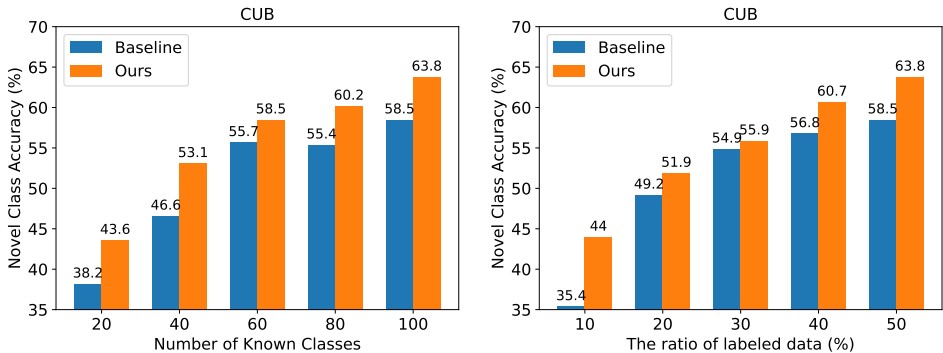

Figure 7: The left and right plots show the model performance with fewer known classes and lower labeling ratios, respectively.

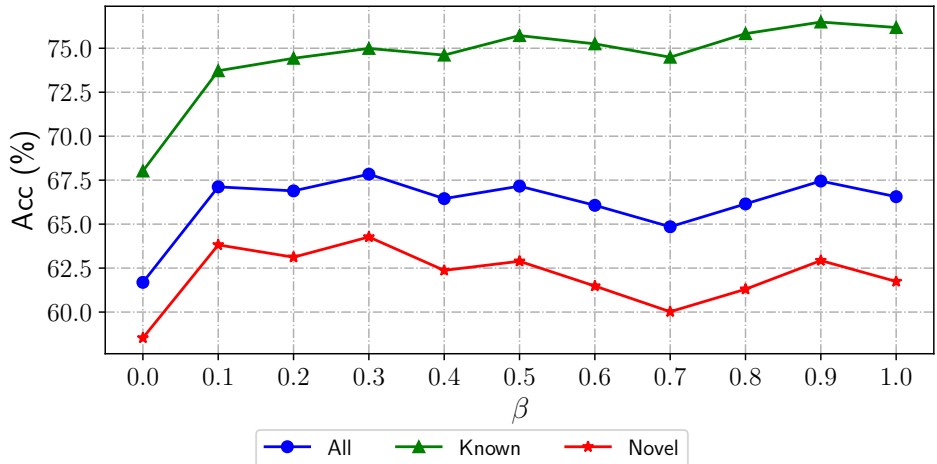

Figure 8: Analysis of hyperparameter $\beta$ on CUB.

# M    SENSITIVE ANALYSIS OF HYPERPARAMETER $\beta$

In our approach, we introduce a hyperparameter $\beta$ to control the strength of representation alignment, as illustrated in Eqn.(5). In Fig.8, we show the model performance with different values of $\beta$. Compared to the baseline model ($\beta = 0$), the model with different values of $\beta$ has a significant improvement. This result indicates that our model is robust to the value of $\beta$. Notably, as $\beta$ increases, the model's novel class accuracy decreases, while the known class accuracy increases. Based on our results, we suggest using a default value of $\beta = 0.1$, which can achieve a balance between known and novel class learning.

