# OpenReview forum: "Adaptive Knowledge Transfer for Generalized Category Discovery"
_ICLR.cc/2024/Conference — Submitted to ICLR 2024_

### Official Review · Reviewer_UEjY · 2023-10-27

**Soundness:** 3 good
**Presentation:** 3 good
**Contribution:** 2 fair
**Rating:** 5
**Confidence:** 5

**Summary:**

This paper targets the task of generalized class discovery (GCD), and argues that the explicit knowledge transfer is a necessity that is largely ignored by existing works. To this end, the authors propose to achieve this goal with three steps: 1) knowledge generation, 2) knowledge alignment and 3) knowledge distillation, which are implemented in a two-stage training procedure. The authors conduct extensive experimental evaluations, which demonstrate that the proposed method largely outperforms existing works.

**Strengths:**

- The paper is well organized and clearly written.
- The experiments on six existing benchmarks and a newly introduced iNat21 are extensive.
- The experimental results showcase the superiority of the proposed method.

**Weaknesses:**

- I believe the idea of knowledge utilization is good, but the proposed terms seem a bit of "big" to me. Technically speaking, knowledge generation, alignment and distillation are respectively training on labeled data, filtering out the important feature dimensions and applying contrastive loss. If there is no obvious significance, I would suggest using more focused terms that better convey the precise purpose.

- As far as I am concerned, the most important component is regarding the "knowledge distillation" (as shown in Tab. 4), where an InfoNCE-like loss is used to do the trick. Within this loss, the MixUp-based negative sample generation seems an important ingredient, yet with no proper ablation studies. Given the potential similarity between the proposed generation procedure and the ones used in [a,b], the authors should provide more theoretical and empirical insights on the difference and significance.

[a] Openmix: Reviving known knowledge for discovering novel visual categories in an open world (CVPR 2021)

[b] Neighborhood contrastive learning for novel class discovery (CVPR 2021)

**Questions:**

In Fig. 6, does the visualizations mean that there are more channels transferred in generic datasets like ImageNet, yet less channels transferred in fine-grained datasets?

---

> ### Author Response · Authors · 2023-11-19
> **Response to Reviewer UEjY**
>
> We sincerely appreciate the thorough evaluation and constructive feedback provided. We hope that the following response could address your concerns.
>
> ## Weakness
>
> (1) Thank you for your clarification. We aim to convey a clear overview of explicit knowledge transfer within GCD problems. While we instantiate knowledge generation, alignment, and distillation using a known-class pre-trained model, AL and CSM, and nMI, our framework allows for flexibility and alternative methods can be employed for these components. For instance, as outlined in the paper, the contrastive loss we utilize can be replaced by other knowledge distillation methods. The primary objective of this writing is to elucidate the explicit knowledge transfer framework for GCD problems, highlight the core challenges associated with knowledge transfer in GCD problems, and offer insights for future research.
>
> (2) We respectfully disagree with you on three points:
> 1. Knowledge distillation is not the most important component of our method. It holds equal importance with other components, AL and CSM. The ablation study in Tab.4 demonstrates that each component significantly contributes to the model's improvement. As discussed in the paper, not all knowledge in known class data is beneficial for novel class learning. This indicates that nMI does not always enhance the model's performance, as supported by the evidence in Tab.12 in Appendix G. This underscores the importance of AL and CSM.
> 2. We would like to clarify that we do not claim novelty in negative sample generation within the paper, which is widely used across various contrastive learning domains. All the other negative sample generation strategies that avoid class collision issues like [2, 3] can also be applied to our method.
> 3. Negative sample generation is not an important component in our method. As shown in the table below, the model's performance without the "negative sample generation" technique remains satisfiable. The reason we opted to use "negative sample generation" is to address the class collision issues of novel classes, as evidenced in the presented results.
>
> | Method | CUB | | | StanfordCars | | | Aircraft | | |
> |-------------------------|--------------------------|-----------------------------------|------------------------------|--------------------------|-----------------------------------|------------------------------|--------------------------|-----------------------------------|------------------------------|
> |                         | All                      | Seen                              | Novel                        | All  | Seen | Novel | All  | Seen | Novel |
> | Ours w/o NSG                 | 67.2                     | 73.2                              | 64.2                         | 58.9 | 80.2 | 48.6  | 54.1 | 60.3 | 51.0  |
> | Ours                     | 67.1                     | 73.7                              | 63.8                         | 59.2 | 79.1 | 49.6  | 55.9 | 60.7 | 53.6  |
>
> ## Question
> We would like to clarify that Fig. 6 illustrates the average outcome of the Channel Selection Matrix generated for each sample in the corresponding class. It can well reflect whether CSM can produce a general selection method for a specific class. However, it does not represent the number of open channels. Additionally, it's worth noting that the values represented by the color depth of each subfigure in Fig. 6 differ, with the maximum value of 0.002 on ImageNet being much smaller than observed on several other datasets. Here, we present the average number of open channels on novel classes in the table below, indicating that fewer channels are transferred in generic datasets, while more channels are transferred in fine-grained datasets.
>
> |                                    | CUB                  | StanfordCars         | Aircraft             | CIFAR 100-80         | ImageNet100         |
> |:-----------------------------------:|----------------------|----------------------|----------------------|----------------------|----------------------|
> | #open channels | 316                    | 284                    | 254                    | 224               | 245 |
>
> [1] Openmix: Reviving known knowledge for discovering novel visual categories in an open world (CVPR 2021)
>
> [2] Neighborhood contrastive learning for novel class discovery (CVPR 2021)
>
> [3] Zheng, Mingkai, Fei Wang, Shan You, Chen Qian, Changshui Zhang, Xiaogang Wang, and Chang Xu. "Weakly supervised contrastive learning." In Proceedings of the IEEE/CVF International Conference on Computer Vision, pp. 10042-10051. 2021.

---

### Official Review · Reviewer_ZnW3 · 2023-10-30

**Soundness:** 2 fair
**Presentation:** 3 good
**Contribution:** 2 fair
**Rating:** 5
**Confidence:** 4

**Summary:**

This paper tackles the generalized category discovery problem. Three steps are proposed to improve the performance. The first step is to learn a pre-trained model with known classes. Then an adapter layer and a channel selection matrix are further learned for knowledge transfer. The last step is to use knowledge distillation. The experiments are conducted on conventional benchmarks and a new benchmark with different difficulty levels.

**Strengths:**

(1) It is clearly written and easy to follow.
(2) Table 1 and Figure 1 show clear motivation for the proposed ideas.
(3) The experimental section is comprehensive and the performance gain is significant in some cases.

**Weaknesses:**

(1) The proposed techniques are well-known in the field and I have the feeling that they are not specifically designed for the GCD problem.
(2) The channel selection seems too simple, I am wondering if there are more sophisticated options.
(3) The experimental results on some datasets are much better than the others but worse on CUB, It would be good to discuss the possible reasons why this is happening.

**Questions:**

The contrastive loss used in the paper is quite common and it would be good to further discuss the novelty of the method.

---

> ### Author Response · Authors · 2023-11-19
> **Response to Reviewer ZnW3**
>
> We sincerely appreciate the thorough evaluation and constructive feedback provided. We hope that the following response could address your concerns.
>
> ## Weakness
>
> (1) We would like to clarify the novelty of our method and its close relationship with the GCD problem in three points:
> 1. Our framework represents a novel approach to addressing GCD challenges, specifically designed for explicit knowledge transfer in GCD tasks. To our best knowledge, no similar framework has been proposed previously for GCD tasks. As discussed in the introduction, knowledge generation, knowledge alignment, and knowledge distillation in our framework aim to generate known class knowledge representation, facilitate more targeted knowledge transfer, and perform knowledge transfer, respectively. Each component is specifically designed for the task of explicit knowledge transfer in GCD.
> 2. The techniques proposed in our work are intricately aligned with this innovative framework. Our proposed AL and CSM are unique solutions crafted specifically to tackle knowledge transfer challenges in the GCD problem. AL aligns the known-class representation space to the joint representation space, and CSM selects meaningful channels, enabling more targeted knowledge transfer. To the best of our knowledge, no such techniques have been proposed before.
> 3. We utilize the contrastive loss to maximize the mutual information. The contrastive loss we introduce is not a traditional one; rather, it uniquely combines AL and CSM, tailoring it to the intricacies of the GCD problem. This novel contrastive loss enables effective knowledge transfer between known and novel classes, unlike traditional knowledge transfer methods.
>
> (2) We contend that the use of ReLU activation is a deliberate choice based on its simplicity and efficiency. And we believe that simplicity is not a drawback but rather an advantage, contributing to the ease of implementing our proposed framework and inspiring future work in designing more sophisticated mechanisms for better knowledge selection.
>
> (3) We would like to point out that the goal of GCD is to learn a model, that can better classify known classes and cluster novel classes, concurrently. Since most existing methods can adapt the model's learning focus by adjusting the weight of $\mathcal{L} _ {u}$ in Equ (1), it is not adequate to evaluate the performance of the model based only on a single metric in the GCD problem. A case in point is the CUB dataset, where the DCCL method exhibits superior performance on novel classes compared to all other methods, yet its performance on known classes is inferior. This observation suggests a potential bias in the DCCL method towards novel class data. Unfortunately, comprehensive analysis is difficult as DCCL has not released its code. When considering all metrics comprehensively, our model is also better than other models on CUB.
>
> ## Question
>
> Please refer to the weakness section.

---

> > ### Comment · Reviewer_ZnW3 · 2023-11-22
> >
> > Thanks for your responses.

---

> > > ### Author Response · Authors · 2023-11-22
> > >
> > > Thank you for your reply. Have we addressed your concerns?

---

### Official Review · Reviewer_Npp8 · 2023-10-31

**Soundness:** 3 good
**Presentation:** 3 good
**Contribution:** 2 fair
**Rating:** 6
**Confidence:** 3

**Summary:**

This paper presents a unique adaptive knowledge transfer framework designed for generalized category discovery, aiming to create a clear connection for knowledge transfer between known and novel classes. The framework is divided into three main components: knowledge generation using a model trained on known class data, knowledge alignment with an adapter layer and a channel selection matrix for more precise knowledge transfer, and knowledge distillation to maximize mutual information between two representation spaces.

Extensive evaluations demonstrate the superiority of this approach over existing methods, and the framework provides a new perspective for advancing knowledge transfer in generalized category discovery, showing great potential to address the challenge of transferring knowledge from known to novel classes effectively.

**Strengths:**

1. This paper introduces a novel framework, meticulously deconstructing the GCD task into three pivotal stages: knowledge generation, knowledge alignment, and knowledge distillation. This explicit dissection embeds a layer of human prior knowledge into the design of the network architecture, showcasing an innovative and finely crafted approach.
2. The results across the majority of datasets indicate that the method achieves state-of-the-art performance.

**Weaknesses:**

The ablation study section on the Adapter Layer (AL) contains several perplexing aspects that raise questions about the method's efficacy, generalizability, and the readability of the paper.

1. In Table 4, the inclusion of AL appears to result in a performance decline on the Scars benchmark, yet the authors provide no explanation for this, which makes me question the generalizability of the method.

2. In Table 5, the distinction between AL and ours seems to be inadequately clarified, rendering this part somewhat hard to follow.

**Questions:**

As noted in the Weakness, I wonder why there is a observed decline in results on the Scars dataset following the incorporation of the Adapter Layer (AL). Is this reduction in performance linked to specific characteristics of the dataset itself, or are there elements of the method that may not be as effective when applied to Scars? I am looking forward to the authors' elucidation on this matter.

Additionally, I would like the authors to clarify the distinctions between the rows for AL and ours in Table 5, with a particular emphasis on the columns where the values are 0.

---

> ### Author Response · Authors · 2023-11-19
> **Response to Reviewer Npp8**
>
> We sincerely appreciate the thorough evaluation and constructive feedback provided. We hope that the following response could address your concerns.
>
> (1) Firstly, concerning AL, its objective is to transform the representation space of known classes into a joint representation space that is more advantageous for novel classes. However, because the transformation introduces novel classes, it may have a negative impact on known classes, depending on the relationship between known and novel classes, which is dataset-specific. We further conduct an ablation study on the coarse-grained dataset in Appendix G, where we also observe a performance drop in known classes and an increase in novel classes when applying AL. The potential negative impact of AL underscores the importance of the design of the CSM. It is noteworthy that the features obtained after both AL and CSM can be considered beneficial for novel class learning. This observation is further supported by the results of the ablation study. Secondly, we would like to clarify that our model integrates not only AL but also nMI and CSM. The utilization of all three components collectively results in notable improvements across all known and novel classes in all datasets. This underscores the method's robust generalizability.
>
> (2) Apologies for any confusion. In Table 5, we assess the model with a frozen known-class pre-trained encoder, adapter layer, and cluster head (denoted as 'AL') to evaluate the quality of the Adapter Layer. The adapter layer and cluster head are learned by $\mathcal{L} _ {base}$. When the number of layers is set to 0, it means we only learn a cluster head. "Ours" denotes our final model with all components. Importantly, when the number of layers is set to 0, "Ours" directly applies nMI without AL and CSM modules.

---

### Official Review · Reviewer_tjGj · 2023-10-31

**Soundness:** 3 good
**Presentation:** 2 fair
**Contribution:** 2 fair
**Rating:** 5
**Confidence:** 4

**Summary:**

This paper addresses the challenge of generalized category discovery, which involves the identification of novel classes in unlabeled data by leveraging information from known classes. Existing methods typically employ a shared encoder to transfer information between labeled and unlabeled data. In contrast, this paper presents a novel framework for explicit knowledge distillation, ensuring effective knowledge transfer from labeled to unlabeled data. The framework incorporates an adaptive layer, a channel-wise selection matrix, and a naive mutual information loss. Additionally, the paper introduces a benchmark dataset called iNat21, along with several data split schemes that consider the semantic gap between labeled and unlabeled data. Experimental results demonstrate the superior performance of the proposed method compared to existing state-of-the-art approaches.

**Strengths:**

(1) This method demonstrates consistent effectiveness across multiple datasets and split schemes, surpassing the performance of previous state-of-the-art approaches.

(2) The main contribution of this work is the introduction of a channel-wise selection matrix, which plays a crucial role in facilitating effective knowledge distillation.

(3) Furthermore, the proposed framework incorporates a naive Mutual Information loss, which is compatible with various formats such as InfoNCE loss, MSE loss, and KL divergence. This compatibility enhances the flexibility and applicability of the framework in different scenarios.

**Weaknesses:**

(1) Some experiments and tables in this paper lack explanation and details. For example:
a) The "BL" and "Clu" settings in Table 1 require further clarification.
b) The specification of the baseline model used in the ablation study needs to be provided.
c) Implementation details of crNCD are missing.

(2) The key idea of knowledge distillation under the NCD setting has been explored in the literature, for example [1, 2], while discussion and analysis are missing.

(3) The paper lacks an explanation of how the naive Mutual Information loss is derived in the format of InfoNCE from the objective of mutual information.

(4) Herbarium is also a commonly used dataset to evaluate the performance of GCD. However, this is missing in the experiments.

[1] Zhao et al, Novel Visual Category Discovery with Dual Ranking Statistics and Mutual Knowledge Distillation, NeurIPS 2021
[2] Gu et al. Class-relation Knowledge Distillation for Novel Class Discovery, ICCV 2023

**Questions:**

(1) The paper mentions training the model for 20 epochs on labeled data in the first stage, but it lacks an explanation for choosing this specific number. It would be beneficial to provide a rationale for selecting 20 epochs and elaborate on how this choice impacts the results.

(2) In the ablation study concerning the adapter layer, each layer is composed of a linear layer and a ReLU layer. It would be helpful to include an experiment using only one ReLU layer to assess the impact of this specific component.

(3) Table 4 exclusively utilizes fine-grained datasets. To provide a more comprehensive evaluation of the proposed method, it would be also important to analyze each component on a coarse-grained dataset.

(4) Previous work [3] suggests that supervised knowledge is beneficial for the task of NCD when the semantics of labeled and unlabeled data are similar. However, this paper demonstrates that even under a data split where the semantic gap between labeled and unlabeled data is significant, knowledge from a fixed labeled encoder remains helpful. It would be helpful to demonstrate how labeled knowledge transfers to unlabeled data across different scales of semantic differences.

(5) The paper mentions the usage of techniques such as dynamic conception generation [4] and cluster size regularization [5]. Analyzing the effects of these techniques would further reveal contributions to the overall performance of the proposed method.

[3] Li, Ziyun, et al. "Supervised Knowledge May Hurt Novel Class Discovery Performance." arXiv preprint arXiv:2306.03648 (2023).

[4] Pu, Nan, Zhun Zhong, and Nicu Sebe. "Dynamic Conceptional Contrastive Learning for Generalized Category Discovery." Proceedings of the IEEE/CVF Conference on Computer Vision and Pattern Recognition. 2023.

[5] Wen, Xin, Bingchen Zhao, and Xiaojuan Qi. "Parametric classification for generalized category discovery: A baseline study." Proceedings of the IEEE/CVF International Conference on Computer Vision. 2023.

---

> ### Author Response · Authors · 2023-11-19
> **Response to Reviewer tjGj**
>
> We sincerely appreciate the thorough evaluation and constructive feedback provided. We hope that the following response could address your concerns.
>
> ## Weakness
> (1)  We appreciate your feedback and have addressed the issues in the revised version of the paper.
>
> - “BL” is the baseline model. In detail, the baseline model has the same architecture as the final model and only uses $\mathcal{L} _ {base}$ in Equ (1) to learn known and novel classes. The loss can influence both the encoder and the classifier. Meanwhile,  “Clu.” denotes that we directly cluster the features generated by the pre-trained known-class model. The model of “Clu.” is also the model with the same architecture as the final model and uses the same $\mathcal{L} _ {base}$ to learn known and novel classes. But now, the encoder is frozen, meaning the loss can only influence the classifier. In Appendix H, we also discuss the details of clustering.
> - This baseline model is the same model “BL” that we discussed above.
> - As crNCD is originally designed for NCD problems, we have adapted it to the GCD setting using a widely employed architecture utilized in other GCD methods. Additionally, to mitigate the influence of different self-labeling designs, we employ the same $\mathcal{L} _ {base}$ (Equ.1) as in our method, instead of using the original $\mathcal{L} _ {opt}$ from crNCD. The original $\mathcal{L} _ {opt}$ in crNCD did not perform well in the GCD setting, as indicated in the table below. It is important to note that we have made no modifications to the core loss $\mathcal{L} _ {rKT}$.
>
> | Method         | CUB | | | StanfordCars | | | Aircraft | | |
> |---------------------------------|--------------------------|-----------------------------------|------------------------------|--------------------------|-----------------------------------|------------------------------|--------------------------|-----------------------------------|------------------------------|
> |                                 | All                      | Seen                              | Novel                        | All  | Seen | Novel | All  | Seen | Novel |
> | $\mathcal{L} _ {opt}$    + crNCD | 56.9                     | 57.8                              | 56.4                         | 46.6 | 62.3 | 39.1  | 45.2 | 59.0 | 43.4  |
> | $\mathcal{L} _ {base}$ + crNCD        | 64.1                     | 75.2                              | 58.6                         | 54.8 | 76.5 | 44.3  | 53.1 | 57.0 | 51.3  |
>
>
> (2) We would like to clarify the distinctions between our method and those proposed in [1,2]. We note that a detailed discussion of [2] has been provided in the related work section.
>
> - Here we summarize the three main differences between our methods and [2] for clarity.
>   1. Our paper proposes a novel explicit knowledge transfer framework that introduces a fresh perspective for advancing knowledge transfer and [2] is only one of the concrete implementations of our proposed framework. Compared with the limited form of knowledge transfer in [2], our framework is more flexible and effective.
>   2. [2] performs knowledge distillation on the output space, while we conduct it in the feature space which contains more information. [2] does not employ knowledge selection to filter out harmful knowledge or knowledge alignment to transfer knowledge more efficiently; instead, they use a simple weight function to amplify knowledge distillation when class relations are strong.
>   3. [2] is specifically designed for the NCD problem and it achieves poor results in GCD (table above).
> - Regarding [1], the primary objective of their loss function $\mathcal{L} _ {sKLD}$ is to enable the model to capture both global and local information within an image. In contrast, our method is specifically designed to facilitate the transfer of known class knowledge to novel class learning. Notably, [1] employs a two-branch learning framework, with one branch focusing on local part-level information and the other on overall characteristics. They transfer knowledge between these two branches. They do not require knowledge alignment because they perform knowledge distillation in the same label space. In contrast, our approach employs an upper branch dedicated to extracting valuable known class knowledge and proposes AL and CSM to do effective knowledge transfer in different representation spaces. These fundamental differences in the purpose and methods distinguish our work from [1].
>
>
> (3) Thank you for your suggestion. In response, we provide the derivation of infoNCE from mutual information and the details can be found in Appendix A.

---

> > ### Author Response · Authors · 2023-11-19
> > **Response to Reviewer tjGj**
> >
> > (4) We supplement the results on the Herbarium19 dataset, which is a long-tailed dataset. The results show we achieve comparable results on the Herbarium19 with competitive SimGCD, while PromptCAL, DCCL, and crNCD do not report results on the Herbarium19 dataset.
> >
> > | Method | Herbarium | | |
> > |-------------------------|-------------------------------|-------------------------|-------------------------------|
> > |                         | All                           | Seen | Novel |
> > | K-means                 | 12.9                          | 12.9 | 12.8  |
> > | Rankstats               | 27.9                          | 55.8 | 12.8  |
> > | UNO                     | 28.3                          | 53.7 | 14.7  |
> > | ORCA                    | 20.9                          | 30.9 | 15.5  |
> > | GCD                     | 35.4                          | 51   | 27    |
> > | OpenCons                | 39.3                          | 58.9 | 28.6  |
> > | PromptCAL               | -                             | -    | -     |
> > | DCCL                    | -                             | -    | -     |
> > | SimGCD                  | 43.3                          | 57.9 | 35.3  |
> > | crNCD                   | -                             | -    | -     |
> > | Ours                    | 43.0                          | 56.2 | 35.9  |
> >
> >
> > ## Question
> > （1） We employ the "early stopping" technique and have observed that the model's loss and accuracy on the validation set stabilize after 20 epochs. Here the model is the known-class pre-trained model.
> >
> > （2） Thank you for your suggestion. We add it in the revised version of the paper. As shown in table below, if we use only a ReLU layer in the adapter layer, the model will experience a significant drop in all datasets. The decline can be attributed to the inherent limitation of a single ReLU layer, which lacks the capacity required for meaningful feature transformation, ultimately resulting in information loss. This observation emphasizes the critical role of transforming the original representation space into a joint representation space, highlighting the importance of the adapter layer in extracting meaningful information.
> >
> > | Method | CUB | | | StanfordCars | | | Aircraft | | |
> > |-------------------------|--------------------------|-----------------------------------|------------------------------|--------------------------|-----------------------------------|------------------------------|--------------------------|-----------------------------------|------------------------------|
> > |                         | All                      | Seen                              | Novel                        | All  | Seen | Novel | All  | Seen | Novel |
> > | ReLU                    | 65.9                     | 73.3                              | 62.1                         | 56.4 | 76.0 | 47.0  | 54.7 | 62.6 | 50.7  |
> > | ReLU + Linear           | 67.1                     | 73.7                              | 63.8                         | 59.2 | 79.1 | 49.56 | 55.9 | 60.7 | 53.6  |
> >
> >
> >
> > （3）We appreciate your suggestion and have appended the ablation on the coarse-grained dataset to Appendix G. The observation in the table below regarding the coarse-grained dataset reveals a significant drop in novel class performance after using nMI. We attribute this to two main reasons:
> >
> > - The representation space initialized with known classes has not encountered novel classes during training, potentially introducing bias in the final knowledge space towards known classes. This motivated the design of AL in our approach.
> >
> > - The knowledge in known class data is not universally useful, and some information may even be harmful to novel class learning. This phenomenon is more pronounced in coarse-grained datasets. Consequently, we introduce CSM to address this issue by filtering out harmful knowledge. We provide a detailed analysis of this phenomenon in the subsequent section.
> >
> > The table results highlight the limitations of directly using nMI, particularly in scenarios with a large semantic gap between classes. The subsequent improvement seen after employing AL and CSM underscores their efficacy in enabling more targeted knowledge transfer for novel class learning.
> >
> > | nMI | AL | CSM | CIFAR100-80 | | |
> > |----------------------|---------------------|----------------------|-------------------------|-------------------------|-------------------------|
> > |                      |                     |                      | All                     | Known | Novel |
> > |                      |                     |                      | 79.2                    | 78.7  | 80.0  |
> > |    &#10004;       |                     |                      | 79.9                    | 84.6  | 70.5  |
> > | &#10004;          | &#10004;         |                      | 82.2                    | 84.1  | 78.3  |
> > | &#10004;          | &#10004;         | &#10004;          | 82.8                    | 84.0  | 80.3  |

---

> ### Author Response · Authors · 2023-11-19
> **Response to Reviewer tjGj**
>
> (4) Firstly, as evidenced in the table above, directly performing knowledge transfer without AL and CSM may not consistently enhance model performance. This phenomenon confirms that not all supervised knowledge is beneficial. This reaffirms the necessity of introducing AL and CSM to filter and refine the transferred knowledge. Secondly, [3] only do experiments on CIFAR and ImageNet datasets. We follow their method and extend their experiments to iNat21 datasets and find that their conclusions may not hold for iNat21 datasets, emphasizing the dataset-specific nature of their findings. As shown in the table below, the novel class performance of UNO is consistent with our assumption on the data splits, and self-supervised uno, which abandons supervised knowledge, achieves inferior results. The results indicate that
> 1. Supervised knowledge is still beneficial in the hard split of the iNat21 dataset.
> 2. completely eliminating supervised information is not an optimal choice thus emphasizing the importance of transforming and filtering beneficial components in our proposed knowledge transfer framework.
>
> | Method       | fine grain | easy | medium | hard |
> |--------------|------------|------|--------|------|
> | UNO          | 52.3       | 48.6 | 46.4   | 43.1 |
> | Self-sup UNO | 32.6       | 31.1 | 30.3   | 27.6 |
>
>
>
> (5) Firstly, we have to clarify that we do not utilize the "dynamic conception generation"[10] technique. Secondly, the cluster size regularization technique is widely employed in unsupervised learning[5, 6] and GCD[7, 8, 9], and it is not our contribution. It ensures meaningful clustering by preventing the model from grouping all data into a single cluster. Therefore, without cluster size regularization, our model and most other GCD models will decrease greatly. Here, we also present the model's performance without cluster size regularization (CSR) in the table below. This table confirms the importance of cluster size regularization as discussed earlier. Nevertheless, our method can still contribute to performance improvement even without cluster size regularization.
>
> | Method | CUB | | | StanfordCars | | | Aircraft | | |
> |-------------------------|--------------------------|-----------------------------------|------------------------------|--------------------------|-----------------------------------|------------------------------|--------------------------|-----------------------------------|------------------------------|
> |                         | All                      | Seen                              | Novel                        | All  | Seen | Novel | All  | Seen | Novel |
> | Baseline w/o CSR    | 39.6  | 47.6  | 35.6  | 33.2  | 59.3  | 20.6  | 35.6  | 37.5  | 34.7 |
> | Ours w/o CSR  | 41.4  | 45.9  | 39.2  | 37.6  | 55.6  | 28.8  | 37.2  | 37.7  | 36.9 |
> | Baseline            | 61.7  | 68.0  | 58.5  | 49.6  | 56.3  | 46.2  | 51.8  | 71.9  | 42.0 |
> | Ours          | 66.8  | 75.6  | 62.5  | 55.6  | 60.5  | 53.1  | 57.6  | 75.9  | 48.8 |
>
>
> [1] Zhao et al, "Novel Visual Category Discovery with Dual Ranking Statistics and Mutual Knowledge Distillation", NeurIPS 2021.
>
> [2] Gu et al. "Class-relation Knowledge Distillation for Novel Class Discovery", ICCV 2023
>
> [3] Li, Ziyun, et al. "Supervised Knowledge May Hurt Novel Class Discovery Performance." arXiv preprint arXiv:2306.03648 (2023).
>
> [4] Oord AV, Li Y, Vinyals O. "Representation learning with contrastive predictive coding." arXiv preprint arXiv:1807.03748. 2018 Jul 10
>
> [5] Asano YM., Rupprecht C., and Vedaldi A. "Self-labelling via simultaneous clustering and representation learning." In Proc. ICLR, 2020.
>
> [6] Wouter Van Gansbeke, Simon Vandenhende, Stamatios Georgoulis, Marc Proesmans, and Luc Van Gool. "Scan: Learning to classify images without labels." In Proc. ECCV, 2020.
>
> [7] Fini, Enrico, et al. "A unified objective for novel class discovery." Proceedings of the IEEE/CVF International Conference on Computer Vision. 2021.
>
> [8] Wen, Xin, Bingchen Zhao, and Xiaojuan Qi. "Parametric classification for generalized category discovery: A baseline study." Proceedings of the IEEE/CVF International Conference on Computer Vision. 2023.
>
> [9] Kai Han, Sylvestre-Alvise Rebuffi, Sebastien Ehrhardt, Andrea Vedaldi, and Andrew Zisserman.
> Autonovel: Automatically discovering and learning novel visual categories. IEEE Transactions on
> Pattern Analysis and Machine Intelligence, 2021.
>
> [10] Pu, Nan, Zhun Zhong, and Nicu Sebe. "Dynamic Conceptional Contrastive Learning for Generalized Category Discovery." Proceedings of the IEEE/CVF Conference on Computer Vision and Pattern Recognition. 2023.

---

### Official Review · Reviewer_2BvT · 2023-11-01

**Soundness:** 3 good
**Presentation:** 4 excellent
**Contribution:** 3 good
**Rating:** 8
**Confidence:** 4

**Summary:**

This paper addresses the challenge of generalized category discovery (GCD). Unlike prior methods that rely on implicit knowledge transfer via shared representation spaces, this work introduces a framework for explicit and adaptive knowledge transfer to facilitate the discovery of novel classes. The proposed method consists of three main steps: (1) capturing known class knowledge through a pre-trained model, (2) transforming this knowledge via an adapter layer and a channel selection matrix for more effective transfer, and (3) employing knowledge distillation to maximize mutual information between representation spaces. It also presents a new benchmark, iNat21, designed with varying levels of difficulty to assess GCD methods.

**Strengths:**

1- It introduces an innovative solution to the complex issue of generalized category discovery (GCD)

2- The paper is articulate and well-structured, providing clear explanations for the motivations behind the approach and the intricacies of the loss functions used.

3-The literature review is exhaustive, offering a thorough overview of related work in the area.

4- Ablation studies and experiments are rigorously conducted, encompassing a broad spectrum of the method's components, which solidifies the validity of the research.

5-The introduction of the iNat21 benchmark is a valuable asset that will likely drive and shape future research in GCD.

**Weaknesses:**

A potential weakness of the paper could be the use of ReLU activations, which are known for their "dead neuron" issue, potentially leading to some neurons becoming inactive due to poor initialization. This characteristic of ReLU could result in the unintentional filtering out of certain channels that might otherwise be useful for learning a more robust embedding space.

**Questions:**

1-How would the model's performance be impacted if activation functions other than ReLU, such as GeLU, were utilized?

2-Regarding Figure 2, is the 'Cls' in the lower branch designed exclusively for labeled samples? Additionally, in the upper branch, what mechanism allows the model to generate categories for the unknown classes after the classification step?

---

> ### Author Response · Authors · 2023-11-19
> **Response to Reviewer 2BvT**
>
> We sincerely appreciate the thorough evaluation and constructive feedback provided. We hope that our response could address your concerns.
>
> ## Weakness
> We opt for the ReLU activation function due to its simplicity and efficiency. Following your insights, we have conducted an analysis to count the number of dead neurons in the model, and the results are presented in the table below. Fortunately, our final model utilizing ReLU does not exhibit this issue. We believe this is potentially due to our truncated norm initialization. In the subsequent discussion, we will delve into a comparative analysis of the model's performances when employing ReLU and GeLU.
>
> |                                    | CUB                  | StanfordCars         | Aircraft             | CIFAR 100-80         |
> |:-----------------------------------:|----------------------|----------------------|----------------------|----------------------|
> | #dead neurons | 0                    | 0                    | 0                    | 0               |
>
> ## Question
> (1) We appreciate your attention to the activation function choice. As depicted in the table below, adopting GeLU resulted in marginal changes in model performance. These experimental findings indicate that both GeLU and ReLU deliver comparable performances, providing evidence that there are few or no dead neurons in our model. Combining this with the empirical results mentioned above, we maintain that ReLU remains a straightforward and effective choice under a good initialization.
>
> | Method | CUB | | | StanfordCars | | | Aircraft | | |
> |-------------------------|--------------------------|-----------------------------------|------------------------------|--------------------------|-----------------------------------|------------------------------|--------------------------|-----------------------------------|------------------------------|
> |                         | All                      | Seen                              | Novel                        | All  | Seen | Novel | All  | Seen | Novel |
> | GeLU                    | 66.3                     | 73.8                              | 62.5                         | 58.3 | 77.6 | 49.0  | 56.2 | 61.0 | 53.8  |
> | ReLU                    | 67.1                     | 73.7                              | 63.8                         | 59.2 | 79.1 | 49.6  | 55.9 | 60.7 | 53.6  |
>
>
>
> (2) In Figure 2, the 'Cls' in the lower branch and the upper branch both can classify labeled and unlabeled data. The 'Cls' in the upper branch is learned based on $\mathcal{L} _ {base} $  in Equ (1). The loss term $\mathcal{L} _ {u} $ in $\mathcal{L} _ {base} $ is the self-labeling loss. Specifically, it utilizes the predictions of 'Cls' in one view of the image to generate pseudo-labels for the other view of the same image. This mechanism enables the model to assign categories to unlabeled data. As a result, the 'Cls' component can classify both known and novel classes. For a more in-depth explanation of $\mathcal{L} _ {u}$, we invite you to refer to Appendix D. Your interest is greatly appreciated.

---

> > ### Comment · Reviewer_2BvT · 2023-11-22
> >
> > Thank you for your response and for elucidating the concepts discussed in the paper. The suggestion to investigate the impact of actively filtering this vector on overall performance presents an intriguing direction for future research. I appreciate the paper's detailed and engaging examination of the GCD problem. Based on these considerations, I have decided to maintain my original review score.

---

### Meta-Review · Area_Chair_KGgJ · 2023-12-02

**Metareview:**

This paper addresses problem of generalized category discovery (GCD). Specifically, this work introduces a framework for explicit and adaptive knowledge transfer to facilitate the discovery of novel classes. To facilitate the study of GCD, a new benchmark, iNat21 is designed. Experiments on several benchmarks demonstrate the benefit of the proposed method.

**Strengths**

- The paper is well-written and well-structured, providing clear explanations for the motivations behind the approach and the intricacies of the loss functions used.

- The proposed method is effective, achieving consistent improvements on the evaluated datasets.

- A new benchmark, iNat21 is designed

- Ablation studies and experiments are rigorously conducted


**Weaknesses**

- The main concern is the limited novelty. Specifically, the method doesn't have much technical novelty and uses knowledge distillation similar to prior NCD works.

- The contribution is limited to the GCD community: the newly introduced iNat21 could be beneficial for later research but it is not a strong contribution as it is similar to existing SSB datasets.

**Justification For Why Not Higher Score:**

The novelty is limited compared to previous methods and the contribution to the community is limited as it seems like yet another method for GCD, which does not bring too much exciting or surprising and may not well-meet the novel requirement of ICLR.

**Justification For Why Not Lower Score:**

N/A

---

### Decision · Program_Chairs · 2024-01-16

Reject